# An atlas of O-linked glycosylation on peptide hormones reveals diverse biological roles

Thomas D. Madsen [1], Lasse H. Hansen [1,2], John Hintze [1], Zilu Ye [1], Shifa Jebari[3], Daniel B. Andersen[4,5], Hiren J. Joshi [1], Tongzhong Ju[6], Jens P. Goetze[2,4], Cesar Martin [6], Mette M. Rosenkilde [4], Jens J. Holst [4,5], Rune E. Kuhre[4,5], Christoffer K. Goth [1], Sergey Y. Vakhrushev [1] & Katrine T. Schjoldager [1]✉

Peptide hormones and neuropeptides encompass a large class of bioactive peptides that regulate physiological processes like anxiety, blood glucose, appetite, inflammation and blood pressure. Here, we execute a focused discovery strategy to provide an extensive map of O-glycans on peptide hormones. We find that almost one third of the 279 classified peptide hormones carry O-glycans. Many of the identified O-glycosites are conserved and are predicted to serve roles in proprotein processing, receptor interaction, biodistribution and biostability. We demonstrate that O-glycans positioned within the receptor binding motifs of members of the neuropeptide Y and glucagon families modulate receptor activation properties and substantially extend peptide half-lives. Our study highlights the importance of O-glycosylation in the biology of peptide hormones, and our map of O-glycosites in this large class of biomolecules serves as a discovery platform for an important class of molecules with potential opportunities for drug designs.

[1] Copenhagen Center for Glycomics, Department of Cellular and Molecular Medicine, Faculty of Health and Medical Sciences, University of Copenhagen, Blegdamsvej 3 DK-2200 Copenhagen N, Denmark. [2] Department of Clinical Biochemistry, Rigshospitalet, University of Copenhagen, Blegdamsvej 9 DK-2100 Copenhagen O, Denmark. [3] Biofisika Institute (UPV/EHU, CSIC), Departamento de Bioquímica, Universidad del País Vasco, Bilbao 48080, Spain. [4] Department of Biomedical Sciences, Faculty of Health and Medical Sciences, University of Copenhagen, Blegdamsvej 3 DK-2200 Copenhagen N, Denmark. [5] Novo Nordisk Foundation Center for Basic Metabolic Research, Faculty of Health and Medical Sciences, University of Copenhagen, Blegdamsvej 3 DK-2200 Copenhagen N, Denmark. [6] Office of Biotechnology Products, Center for Drug Evaluation and Research, U.S. Food and Drug Administration, Silver Spring, MD 20993, USA. ✉email: Schjoldager@sund.ku.dk

Peptide hormones, neuropeptides, and other biologically active peptides represent one of the largest and most promising classes of drug candidates. A wide range of synthetic analogs are emerging as major drug candidates for treatment of neurological and metabolic disorders, mediating their physiological effects by acting as either agonists or as antagonists on their cognate receptors[1]. While peptide hormone-based drugs benefit from being selective for their cognate receptors, the usage of native peptides as therapeutics is challenging due to the inherent instability giving rise to short half-lifes of the biologically active peptide form(s) on a scale of a few minutes[2]. Thus, considerable efforts are being devoted to discovery and development of peptide-based analogs to improve stability and circulation time, as well as safety, immune tolerance, efficacy, and biodistribution. Different strategies include use of unnatural amino acids, modifications such as lipidation, PEGylation, conjugation to antibodies, and stapled peptides[3]. A prerequisite for rational design and development of improved peptide hormone drugs is knowledge of the molecular composition and structure of the natural bioactive peptides, and their mode of interaction with receptors.

Peptide hormones are synthesized as precursor proteins that may undergo regulated proteolytic processing during their transport through the secretory pathway to become mature hormones (involving enzymes such as carboxypeptidases[4], proprotein convertases (PCs)[5], and C-terminal α-amidation[6]). In addition, a number of other posttranslational modifications (PTMs) have been identified, including tyrosine sulfation[7], N-terminal acetylation[8], phosphorylation[9], and glycosylation[10]. Unlike constitutively secreted proteins, most peptide hormones are stored within the cell. Upon secretion, most peptide hormones are prone to specific and rapid proteolytic degradation resulting in short circulatory half-lives[11–13]. As a result, the plasma concentration of most peptide hormones is in the lower picomolar range, which for technical reasons has challenged the peptidomic insight into the molecular composition of the mature forms of these hormones and their PTMs. Recent sensitive mass spectrometry (MS)-based studies have identified alpha-amidation, acetylation, and phosphorylation on select peptide hormones[9,14]. However, given that the vast majority of proteins trafficking the secretory pathway undergo O-glycosylation, it is surprising that only very few O-glycosylated peptide hormones have been identified to date[10,15–19].

Mucin-type O-glycosylation (hereafter O-glycosylation) is a highly abundant PTM estimated to affect 80% of secreted proteins in a cell[15]. The biosynthesis is initiated by a large family of 20 partly redundant polypeptide GalNAc-transferases (GalNAc-T) isoenzymes, and recently we and others have demonstrated that O-glycosylation plays wide roles in co-regulation of fundamental processes, such as PC processing, ectodomain shedding, and receptor modulation, to fine-tune and diversify biological functions of proteins[20–22]. Most recently, we identified O-glycosites on the atrial natriuretic peptide (ANP) hormone, and found that O-glycans in the receptor-binding region alter the mode of action of ANP in rats, providing improved therapeutic performance[10]. Prompted by this, we hypothesized that O-glycosylation could be a more prevalent PTM of peptide hormones than recognized to date, and that a focused discovery strategy would reveal this.

Here, we present the design and application of a comprehensive discovery program focused at identifying O-glycans on the large class of low molecular weight peptide hormones. We use different enrichment strategies and targeted MS-based analysis of multiple biosources to produce an atlas of O-glycosites on peptide hormones. We identify wide occurrence of O-glycans on select subfamilies of peptide hormones with multiple O-glycosites in the receptor-binding domains conserved among peptide sequences within subfamilies, as well as through evolution. We demonstrate that O-glycans in receptor-binding regions serve to modulate both peptide hormone stability and receptor signaling.

## Results

**Workflow for discovery of glycans on peptide hormones**. We designed a proteomics-based workflow targeting low molecular weight glycoproteins using low molecular weight enrichment (LMWE) and lectin weak affinity chromatography (LWAC) enrichments[15,23], and applied a comprehensive database of neuropeptides and peptide hormones (NeuroPep; http://isyslab.info/NeuroPep/)[24] for informatics. The strategy (presented in Supplementary Fig. 1a) was applied to both cell lines (N2A and STC-1) and organs known to express and/or secrete high levels of diverse peptide hormones, including porcine and rat brain, cerebellum, heart, and pancreas as well as porcine intestinal ileum, and human plasma, cerebrospinal fluid (CSF), and prostate cancer biopsies (Fig. 1).

Across all biosources, we identified ~97,000 peptide-spectrum matches containing glycan information (Supplementary Fig. 1a). In total, 445 O-glycosites were found to be located on distinct prohormone proteins[24]. Of these, 36 sites were identified on human prohormone sequences and 409 glycosites were identified on nonhuman (porcine or rodent) prohormones. The latter were subjected to prediction of conservation in human neuropeptide orthologs by sequence alignment analysis and revealed that 374 out of 409 glycosites were predicted to be conserved in humans, based on conservative preservation of Ser/Thr residues within (±5) amino acid residues[25] (Supplementary Fig. 1b). In total, 410 (36 + 374) conserved O-glycosylation sites were included for further analysis. Surveying the distribution of the identified human and conserved O-glycosites across each peptide hormone precursor revealed that about half (223) of the 410 glycosites were located in the proprotein part, and the other half (187) were located on the mature peptide hormones (Fig. 1a, Table 1). These O-glycosites were distributed on 91 mature peptide hormone sequences out of the total 279 peptide hormones annotated in the comprehensive Neuropep database[24] (including 14 peptide hormones for which we only observed ambiguously assignable glycosites; Fig. 1b, Table 1, Supplementary Fig. 1b, Supplementary Data 1–4). The identification correlated well with the known tissue expression of specific peptide hormones (e.g., insulin and pancreatic polypeptide (PPY) in pancreas). We identified fewest peptide hormones in plasma, and most in brain and intestinal tissues among the different biosources analyzed (Fig. 1c). In general, the LMWE resulted in more identifications of peptide hormone O-glycosites compared to the traditional analysis of total tissue/cell extracts (Rapigest) and biofluids, and in total contributed with >25% (24) additional peptide hormones with O-glycosylation (Fig. 1d). Each LMWE resulted in the identification of a similar number of glycosylated peptide hormones per biosource analyzed (Supplementary Data 2) where subtle qualitative differences were noted. For example, glycosylated insulin-like growth factor II (IGFII) was only identified in CSF after EtOH extraction, glycosylated glucagon-like peptide 1 (GLP-1) was found in ileum only after acidic extraction, and glycosylated CCB peptide was observed only after acetone extraction in both whole brain and cerebellum.

**An atlas of peptide hormone O-glycosylation**. Human peptide hormones are classified into 46 distinct families of which O-glycosylated members were identified in 28 of the families (Fig. 2, Supplementary Data 1–3). In addition, the algorithm NetOGlyc 4.0 (ref. [15]; http://www.cbs.dtu.dk/services/NetOGlyc/) predicted that the majority of peptide hormone families contain

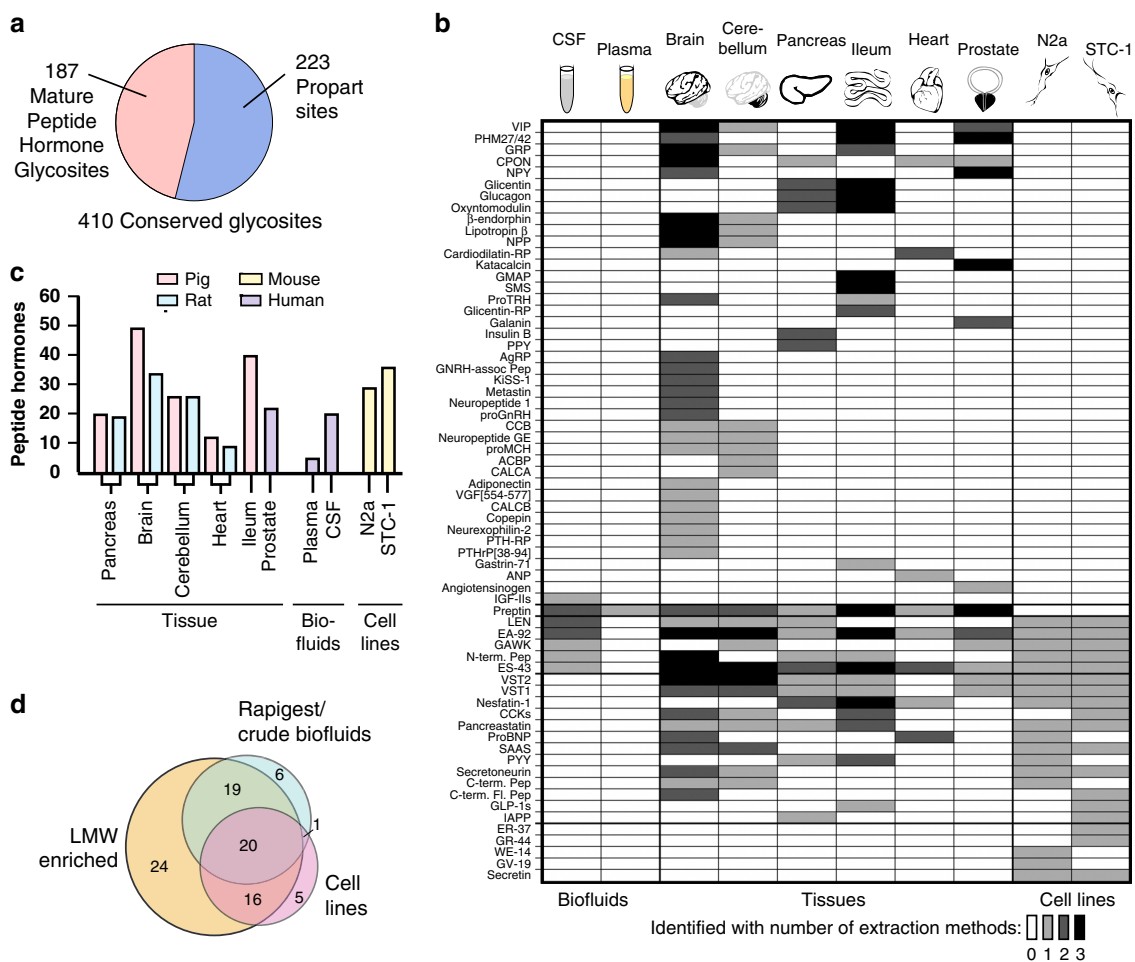

**Fig. 1 Summary of the identification of glycosylated peptide hormones using LC–MS/MS. a** Number of glycosites identified on mature and propeptide hormone proteins. **b** Heatmap illustrating the identification of glycans on mature peptide hormones across different biofluids, tissues, and cell lines. Greyscale intensity illustrates the number of extraction methods that have resulted in the identification of a given glycopeptide. See Supplementary Data 4 for examples of extracted spectra of glycopeptides from each peptide hormone. **c** Bar graph of the total number of mature glycopeptide hormones identified in each tissue or cell line (all enzymatic digestion procedures combined). **d** Venn diagram illustrating the distribution of identified glycosylated peptide hormones across biosources between the different extraction methods (total protein extractions (Rapigest extractions and crude biofluids), low molecular weight extractions (LMWE: H₂O, acetone, acetone/AcOH, AcOH, and EtOH/HCl) and cell lines (supernatant and total cell lysate)). Please see Supplementary Data 1 for peptide hormone abbreviations.

glycosylated members (41 out of 46). Among the glycosylated peptide hormone families, well-characterized peptides like GLP-1, cholecystokinin (CCK), neuropeptide Y (NPY), galanin (GAL), and secretin (SCT) were found glycosylated.

We analyzed glycosite locations relative to protein topology and known functional features and found that the majority of sites were distributed among regions of the peptide hormones typically referred to as the address and message regions[26]. Approximately 10% of the glycosites were located ±5 amino acids from a PC processing site (Fig. 3a). For example, in proopiomelanocortin (POMC; Supplementary Fig. 2a), we confirmed the Thr71 glycosite adjacent to the [71]TENPRK↓Y[77] processing site, which inhibits activation of gamma-melanocyte stimulating hormone[27]. In proNPY (Supplementary Fig. 2b), we identified Ser68 and Ser69 glycosites immediately adjacent to the PC1/3 activation ([63]RYGKR↓SS[69]) site in NPY[28]. In somatostatin (Supplementary Fig. 2c), an ambiguous glycosite (Ser89 or Ser92) was found just C-terminal to the furin/PACE4 activating site [85]ELQR↓SANS[92] [29].

We explored the address and message glycosites relative to characterized structural and/or functional receptor-binding motifs. Interestingly, in five different peptide hormone families

a majority of the glycosites were located in the highly conserved functional domains involved in receptor interaction (Fig. 3b). A detailed description of all five families is included in Supplementary Fig. 3.

**O-glycans on peptide hormones modulate receptor activation.** The glucagon (GCG) family members share a highly conserved O-glycan at Thr7, while the NPY family members share a conserved O-glycan at Thr32 (Fig. 4a, b). To explore the potential functional impact of site-specific O-glycosylation on mature peptide hormones, we chose to study members of these families (vasoactive intestinal polypeptide (VIP), GCG, GLP-1, NPY, and peptide YY (PYY)). We chemoenzymatically synthesized the candidate hormones with the three most common O-GalNAc-type structures, including Tn (GalNAcα1-O-Ser/Thr), T (Galβ1-3GalNAcα1-O-Ser/Thr), and sialylated ST (NeuAcα2-3Galβ1-3GalNAcα1-O-Ser/Thr; Fig. 4c), to generate NPY[Thr32], PYY[Thr32], VIP[Thr7], GCG[Thr7], and GLP-1[Thr7]. These peptide glycovariants were used for classical dose–response studies using COS-7 cells that transiently expressed relevant cognate receptors. All receptors were activated by the relevant non-modified peptide ligands

## Table 1 Atlas of peptide hormone glycosylation.

| Family name | Peptide hormones | Brain pig | Brain rat | Cerebellum pig | Cerebellum rat | Heart pig | Heart rat | Pancreas pig | Pancreas rat | Ileum pig | Prostate human | CSF human | Plasma human | N2a mouse | STC-1 mouse | Phosphorylated[a] |
|---|---|---|---|---|---|---|---|---|---|---|---|---|---|---|---|---|
| 7B2 | C-terminal peptide | X | X | – | – | – | – | – | – | X | – | – | – | X | X | X |
| 7B2 | N-terminal peptide | X | X | X | – | – | – | X | – | X | – | X | – | X | X | X |
| 7B2 | Neuroendocrine protein 7B2 | X | – | X | – | – | – | X | – | X | – | X | – | – | – | X |
| ACBP | Acyl-CoA-binding protein | – | – | – | – | – | – | – | – | – | – | – | – | – | – | – |
| ADH | Copeptin | – | – | – | – | – | – | – | – | – | – | – | – | – | – | – |
| Bombesin | Gastrin-releasing peptide | X | X | X | X | – | – | – | – | X | – | – | – | X | – | – |
| Calcitonin | Calcitonin gene-related peptide 1 | – | – | – | X | – | – | – | – | – | – | – | – | – | – | X |
| Calcitonin | Calcitonin gene-related peptide 2 | X | – | – | X | – | – | – | – | – | X | – | – | – | – | X |
| Calcitonin | Islet amyloid polypeptide | – | – | – | – | – | – | X | X | – | X | – | – | – | X | – |
| Calcitonin | Katacalcin | – | – | – | – | – | – | – | – | – | X | X | X | – | – | – |
| CCKs | Cholecystokinin-25/33/39/58/(desnopeptide) | X | – | X | – | – | – | – | – | X | – | – | – | X | X | X |
| CCKs | Gastrin-71 | – | – | – | – | – | – | – | – | X | – | – | – | – | – | X |
| Granins | CCB peptide | X | X | X | X | X | – | X | X | X | X | X | – | X | X | X |
| Granins | EA-92 | X | X | X | X | X | – | X | X | X | X | X | – | X | X | X |
| Granins | ES-43 | – | – | – | – | – | X | – | – | – | X | X | – | – | – | X |
| Granins | GAWK peptide | X | X | X | X | X | X | X | X | X | X | X | – | – | X | X |
| Granins | GR-44/ER-37 | X | X | X | X | X | – | X | – | – | – | – | – | – | – | X |
| Granins | GV-19 | – | – | – | – | – | – | – | – | – | – | – | – | – | – | X |
| Granins | Pancreastatin | X | X | X | X | X | X | X | X | X | X | X | – | X | X | X |
| Granins | Chromogranin-A | X | X | X | X | X | X | X | X | X | X | X | – | X | X | X |
| Granins | Secretogranin-1 | X | X | X | X | X | X | X | X | X | X | X | – | X | X | X |
| Granins | Secretogranin-2 | X | X | X | X | X | – | X | X | X | X | X | – | X | X | X |
| Granins | Secretogranin-3 | X | X | X | X | X | – | X | X | X | X | X | – | X | X | X |
| Granins | Secretoneurin | X | X | X | X | – | X | X | X | X | X | X | – | X | X | X |
| Granins | Vasostatin-1 | X | X | X | – | – | – | X | X | X | X | – | – | X | X | X |
| Granins | Vasostatin-2 | X | X | X | X | – | – | X | X | X | X | – | – | X | X | – |
| Granins | WE-14 | – | – | – | – | – | – | – | – | – | – | – | – | X | – | – |
| Cystatin | Kininogen-1 | – | – | X | X | X | X | X | X | X | X | X | – | – | – | – |
| Cystatin | Kininogen-1 heavy chain | – | – | – | – | – | – | – | – | – | X | X | X | – | – | – |
| Cystatin | Kininogen-1 light chain | – | X | X | X | – | – | X | X | X | X | X | X | – | – | – |
| Galanin | Galanin | – | – | – | – | – | – | – | – | – | X | – | – | – | – | X |
| Galanin | Galanin message-associated peptide | – | – | – | – | – | – | – | – | X | – | – | – | – | – | – |
| Glucagon | Glicentin-related polypeptide | – | – | – | – | – | – | – | – | X | – | – | – | – | – | – |
| Glucagon | Glucagon-like peptide 1's | X | – | X | – | X | X | X | X | X | X | X | X | X | X | – |
| Glucagon | Glucagon/glicentin | X | X | X | X | X | – | X | – | X | X | – | – | X | – | – |
| Glucagon | Intestinal peptide PHM-27/PHV-42 | X | X | X | X | – | – | X | X | X | X | X | – | X | – | – |
| Glucagon | Oxyntomodulin | X | X | X | – | – | X | X | X | X | X | – | – | X | X | – |
| Glucagon | Secretin | – | – | – | – | – | – | – | – | – | – | – | – | – | – | – |
| Glucagon | Vasoactive intestinal peptide | X | X | X | X | – | – | X | – | X | X | – | – | X | – | X |
| GnRH | GnRH-associated peptide 1 | X | – | – | – | – | – | X | X | X | X | X | – | – | – | – |
| GnRH | Progonadoliberin-1 | X | – | – | X | – | – | X | X | X | X | X | – | – | – | – |
| Insulin | Insulin B chain | – | – | – | – | – | – | X | X | – | – | – | – | – | – | – |
| Insulin | Insulin-like growth factor II/Ala-25 Del | X | X | X | X | X | – | X | – | X | X | X | X | X | X | X |
| Insulin | Preptin | X | – | – | – | – | – | X | – | X | X | X | X | – | – | – |
| KISS1 | Metastasis-suppressor KiSS-1 | X | X | – | – | – | – | – | – | – | – | – | – | – | – | – |
| KISS1 | Metastin | X | X | – | – | – | – | – | – | – | – | – | – | – | – | – |
| MCH | Neuropeptide-glycine-glutamic acid/pro-MCH | – | X | X | X | – | X | X | X | X | X | – | – | X | X | – |
| NA | Adiponectin | X | X | – | – | – | – | X | X | X | – | – | – | – | – | – |
| NA | Agouti-related protein | – | – | – | – | – | – | – | – | – | – | – | – | – | – | – |
| Natriuretic peptide | Atrial natriuretic factor | X | X | X | X | X | X | X | X | X | X | X | – | X | X | X |
| Natriuretic peptide | Cardiodilatin-related peptide | X | X | X | X | X | X | X | X | X | X | X | – | X | X | X |
| Natriuretic peptide | Natriuretic peptides B | X | – | – | – | X | X | – | – | – | – | – | – | – | – | – |
| Neurexophilin | Neurexophilin-2 | X | X | X | – | – | – | – | – | – | – | – | – | – | – | – |
| NPY | C-flanking peptide of NPY | X | X | X | X | X | – | X | X | X | X | X | – | X | X | X |
| NPY | Neuropeptide Y | X | X | X | X | – | X | X | X | X | X | – | – | X | X | X |
| NPY | Pancreatic hormone (PPY) | – | – | – | – | – | – | X | X | – | – | – | – | – | – | – |
| NPY | Peptide YY/(3-36) | – | – | – | – | – | – | X | X | X | X | – | – | – | – | – |
| Nucleobindin | Nesfatin-1 | – | – | – | – | – | – | – | – | – | – | – | – | – | – | – |
| Nucleobindin | Nucleobindin-1 | X | X | X | X | X | X | X | X | X | X | X | X | X | X | X |
| Nucleobindin | Nucleobindin-2 | X | X | X | X | X | X | X | X | X | X | X | – | X | X | X |
| Opioid | Neuropeptide 1 | X | – | X | – | – | – | X | – | – | X | – | – | – | – | X |
| Parathyroid hormone | PTHrP/PTHrP[38–94] | X | X | – | – | – | – | – | – | – | – | – | – | – | – | X |
| POMC | Beta-endorphin/lipotropin beta/NPP | – | X | – | X | – | – | X | X | – | – | X | – | – | – | – |
| ProSAAS | Big LEN/little LEN/big PEN-LEN | – | X | – | X | – | – | X | X | – | – | – | – | X | X | X |
| ProSAAS | Big SAAS/little SAAS | X | – | X | – | – | – | – | – | – | – | X | – | X | X | X |

**Table 1 (continued)**

| Family name | Peptide hormones | Brain pig | Brain rat | Cerebellum pig | Cerebellum rat | Heart pig | Heart rat | Pancreas pig | Pancreas rat | Ileum pig | Prostate human | CSF human | Plasma human | N2a mouse | STC-1 mouse | Phosphorylated[a] |
|---|---|---|---|---|---|---|---|---|---|---|---|---|---|---|---|---|
| ProSAAS | proSAAS | X | X | X | X | – | – | – | X | – | – | X | – | X | X | X |
| Serpin | Angiotensinogen | X | X | – | – | – | – | – | – | – | X | – | – | X | X | – |
| Somatostatin | Somatostatin-28 | – | – | – | – | – | – | – | – | X | X | – | – | – | X | X |
| Tachykinin | C-terminal-flanking peptide | X | – | – | – | – | – | – | – | X | – | – | – | – | – | – |
| TRH | Pro-thyrotropin-releasing hormone | X | X | – | – | – | – | – | – | X | – | – | – | – | – | – |
| VGF | Antimicrobial peptide VGF[554–577] | – | X | X | X | – | – | – | – | – | X | X | – | – | X | – |
| VGF | Neurosecretory protein VGF | X | X | X | X | – | – | – | – | – | X | X | X | – | – | – |

X indicates the identification of peptide hormone glycosylation in a given biosource. – indicates no identification in a given biosource. (See Supplementary Data 1 for full atlas of glycosylated peptide hormones). Please see Supplementary Data 1 for peptide hormone and family abbreviations.
[a]Peptide hormone phosphorylation sites previously identified[9,14] (Supplementary Data 5).

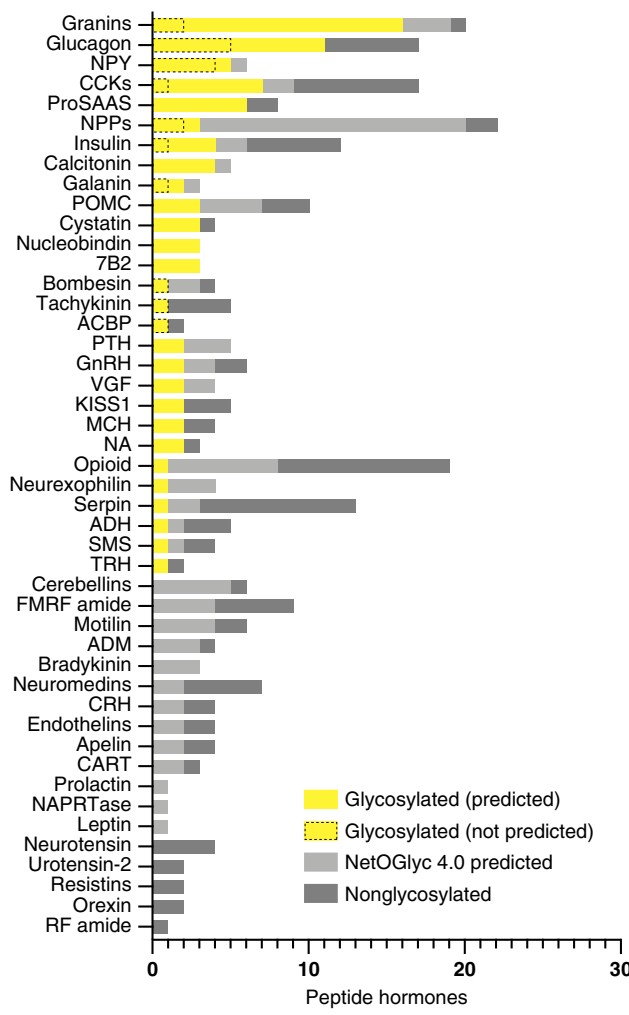

**Fig. 2 Glycosylation of peptide hormones is a widespread phenomenon.** Bar graph illustrating that 28 out of the 46 human families annotated in the NeuroPep[24] database contain at least one glycosylated member (yellow). The glycosylation prediction algorithm, NetOGlyc 4.0, predicts that almost all (41 out of 46) families contain glycopeptide hormones (light gray) and five families only have members without glycosylation (dark gray). The number of identified peptide hormones that NetOGlyc 4.0 does not predict are marked by a dotted line. Please see Supplementary Data 1 for peptide hormone family abbreviations.

producing half the maximal effect ($EC_{50}$) at doses in accordance with the literature[30–33]. A summary of all dose–response curves is presented in Supplementary Fig. 4, and exact $EC_{50}$ values can be found in Supplementary Table 1.

In the GCG family, non-glycosylated VIP exhibited a potency of 0.2 nM and 0.4 nM for VIP receptor 1 and 2 (VPAC1 and 2), respectively. In striking contrast, all three glycoforms of $VIP^{Thr7}$ showed a strongly reduced potency on both VPAC1 and VPAC2 (Fig. 4d). GLP-1 signals exclusively through the GLP-1 receptor (GLP-1R), and in our assay non-glycosylated GLP-1 showed a potency of 0.05 nM for GLP-1R, while $GLP-1^{Thr7}$ produced a less potent yet full maximal response. Activation of the glucagon receptor (GCGR) by GCG occurred with an $EC_{50}$ of 1.3 nM, while $GCG^{Thr7}$ exhibited decreased potency with 100–1000-fold increased $EC_{50}$ dependent on the structure of the installed O-glycan (Fig. 4d, Supplementary Figs. 4 and 5a). The NPY family peptide hormones activate the same members of the Gi-coupled NPY receptor family (NPY1R, NPY2R, NPY4R, and NPY5R; Fig. 4e, Supplementary Fig. 4). In agreement with past studies,

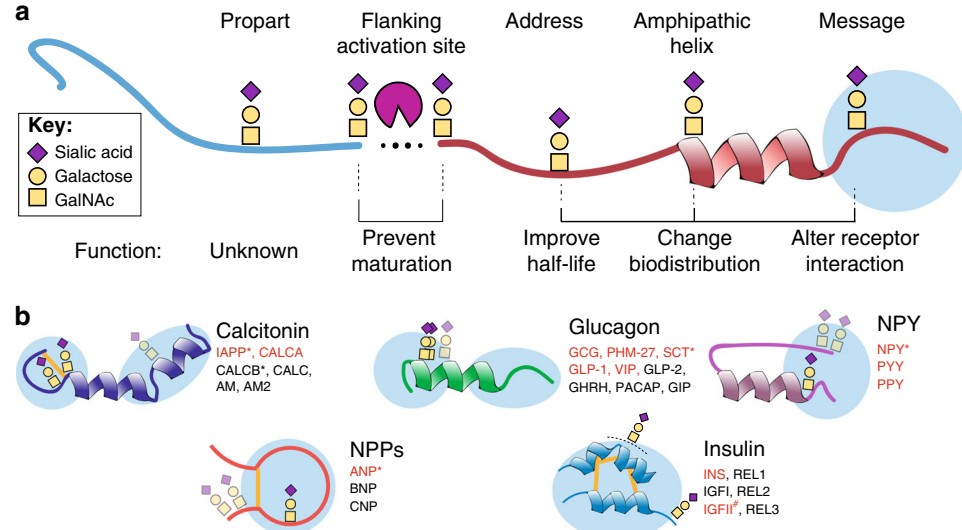

**Fig. 3 Topology of O-glycosites in peptide hormones and their predicted functions. a** Representation of a generic peptide hormone with indications of potential functional impact of glycosylation sites. **b** Illustrations of glycopeptide hormone families with message-associated O-glycosites conserved or semi-conserved among the paralogous family members. The message parts of the peptide hormones are indicated by light blue circles. Glycans have been identified in conserved positions in family members in red. Faded glycosites are not shared within paralogous family members. (*) Denotes peptide hormones containing non-conserved glycosites. (#) The site over the dotted line in the insulin family is covered by ambiguously assigned glycosite(s) in the C-terminal part of IGFII. Full sequence alignments of the five presented glycopeptide hormone families are shown in Supplementary Fig. 3. Please see Supplementary Data 1 for peptide hormone abbreviations.

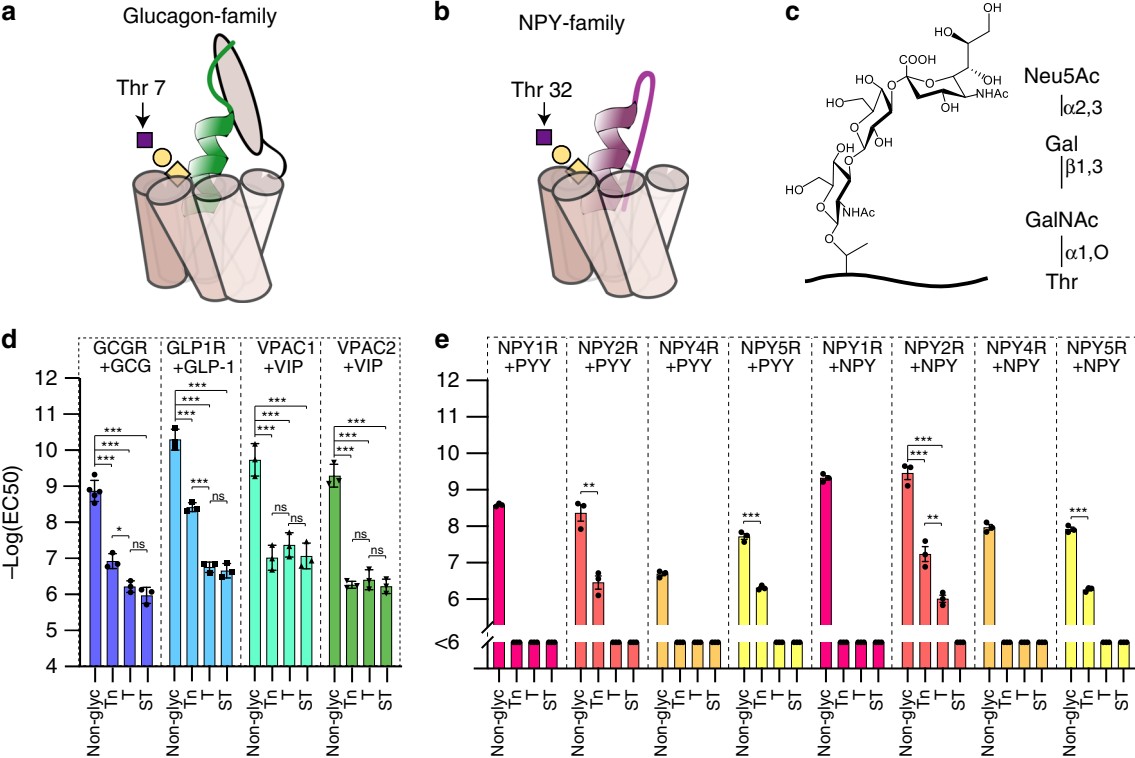

**Fig. 4 Functional analysis of O-glycans within the receptor-binding region of peptide hormones. a** Schematic depiction of the Thr$^7$ O-glycan positioned within the receptor-binding domain of the glucagon hormone family (see Supplementary Fig. 7e). **b** Schematic depiction of the Thr$^{32}$ O-glycan positioned within the receptor-binding domain of the NPY hormone family[38]. **c** Structural representation of the sialyl-T O-glycan. **d** Potencies (Log$_{10}$(EC$_{50}$)) of glycosylated GCG, VIP, and GLP-1 determined in cell-based receptor activation assays ($n = 3$ independent experiments in duplicate determinations for all groups, except non-glyc. GCG where $n = 5$). **e** Potencies of NPY and PYY in their Thr$^{32}$ glycoforms determined and shown as in **d** ($n = 3$ for all groups). All error bars represent mean ± S.E.M. Two-tailed one-way ANOVA (Tukey's post hoc test) was performed in **d** and **e** except for comparisons within NPY2R +PYY-; NPY5R+PYY-, and NPY5R+NPY-assays in **e** where two-tailed Student's t-test was performed. ns not significant; *$p < 0.05$; **$p < 0.01$; ***$p < 0.001$ (all significant p-values are <0.0001 except **d** GCGR-Tn/GCGR-T $p = 0.0149$; **e** NPY2R-PYY-Non-glyc/NPY2R-PYY-Tn $p = 0.0027$; NPY2R-NPY/NPY2R-NPY-Tn $p = 0.0002$; NPY2R-NPY-Tn/NPY2R-NPY-T $p = 0.0046$). Source data for **d** and **e** are provided as a Source data file.

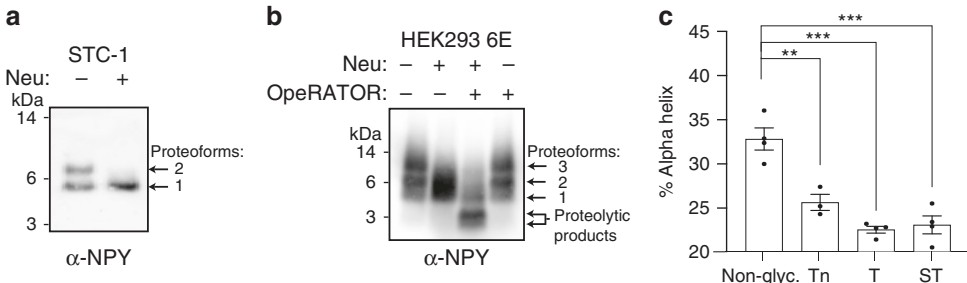

**Fig. 5 Molecular and structural analysis of NPY and its precursor. a** Western blot of conditioned media from STC-1 cells stably expressing proNPY using an antibody recognizing pro- and mature NPY. The conditioned media from proNPY-transfected STC-1 cells was concentrated using TCA precipitation, and analyzed with and without neuraminidase treatment (Neu). **b** Western blot of conditioned media from proNPY-transfected HEK293-6E cells were analyzed with and without neuraminidase treatment (Neu), OpeRATOR treatment, or both in combination. Both western blot experiments in **a** and **b** were repeated three times with similar results. **c** Alpha-helical content of 20 µM NPY or its glycoforms determined by CD spectroscopy (pH 7 at 25 °C) ($n = 4$ independent measurements for all groups except Tn where $n = 3$). All error bars represent mean ± S.E.M. Two-tailed one-way ANOVA (Tukey's post hoc test) was performed in **c**. ***$p < 0.001$, **$p < 0.01$ (all $p$-values are <0.0001 except **c** Non-glyc/Tn where $p = 0.0018$). Source data for **a**–**c** are provided as a Source data file.

non-glycosylated PYY and NPY activated receptors with potencies ($EC_{50}$) between 2.6–204 nM and 0.4–10.7 nM, respectively[34,35]. One exception was activation of the Y5 receptor that showed an $EC_{50}$ value of 204 nM compared to previously reported 1.2 nM[34]. Interestingly, among the four receptors tested, NPY2R and NPY5R partially tolerated $NPY^{Thr32-Tn}$ and $PYY^{Thr32-Tn}$ with $EC_{50}$ values in the range of 58–550 nM (Fig. 4e), whereas NPYR1 and NPYR4 showed only slight activation at the highest dose of the glycosylated peptide hormones tested (Supplementary Figs. 4 and 5b).

In summary, all O-glycosylated peptide hormones elicited a dampened agonist response with increased $EC_{50}$ values correlating positively with the size of the O-glycan. The presence of sialic acid with its strong negative charge, however, did not substantially affect the potency of the members of the GCG family when comparing to the other tested variants. In general, O-glycans located in the receptor-binding domain of the NPY family members had a bigger impact on receptor activation than glycans in the receptor-binding domain of GCG family members (Fig. 4d, e). Further investigation is required to explain this phenomenon, but it could be related to the large extracellular N-terminal domain of the receptors to the GCG family members[36] (like other class B G-protein-coupled receptors), which may allow for binding of the C-terminally non-glycosylated part (Fig. 4a). In contrast, NPY receptors (which belong to A-class G-protein-coupled receptors) do not contain this large extracellular N-terminal domain[37] (Fig. 4b).

**ProNPY is highly glycosylated in a neuroendocrine cell line.** Estimation of stoichiometry using LC–MS/MS is difficult with LWAC-enriched glycopeptide samples, because the non-glycosylated peptide fraction is depleted by the lectin enrichment. In addition to this, the determination of glycosylation stoichiometry in non-enriched samples is challenged by the general sample complexity and dynamic range[38]. Studying the low abundant peptide hormones that are often not detectable in proteomics studies further adds to the difficulties in detection of glycopeptides let alone identification of glycoforms and stoichiometry of glycosylation. Thus, to address stoichiometry with one peptide hormone, we stably expressed full-length preproNPY in the neuroendocrine STC-1 cell line and used traditional SDS–PAGE molecular weight shift assays to evaluate the degree of O-glycosylation. Direct SDS–PAGE western blotting of trichloroacetic acid concentrated conditioned media showed that secreted proNPY migrated as two distinct and equally intense

bands (proteoforms 1 and 2, Fig. 5a, lane 1). Treatment with neuraminidase to remove sialic acids on glycans collapsed the two immunoreactive bands of proNPY into one apparently co-migrating band, which demonstrated that the upper band of proNPY contains sialylated glycans (lane 2). To confirm the presence of GalNAc-type O-glycans, we expressed proNPY in a high producer cell line, HEK293-6E, that secreted proNPY in amounts directly detectable by western blotting without concentrating the sample further. HEK293-6E-produced proNPY migrated as three bands; two upper strong (proteoform 2 and 3) and one lower weak (proteoform 1) band (Fig. 5b, lane 1). As with STC-1, neuraminidase demonstrated that a considerable amount of proNPY contains sialylated glycans (lane 2). For further confirmation, we used a recently characterized O-glycoprotease from *Akkermansia muciniphila*, OpeRATOR, which specifically cleaves O-glycopeptides immediately N-terminal to an O-glycan, but is blocked by the presence of sialic acid[39]. Treatment with OpeRATOR only in combination with neuraminidase resulted in the loss of the intensity of the upper band concomitantly with the appearance of a band migrating approximately as 4–5 kDa (lane 3), confirming that a majority of proNPY contains one or more O-glycans with sialic acid. In our atlas, we identified multiple glycosylation sites on proNPY, and based on this data it was not possible to determine the specific OpeRATOR cleavage site(s) (Supplementary Fig. 6). However, the data confirmed that two mammalian cell lines, STC-1 and HEK293-6E, produce proNPY with at least 50% sialylated GalNAc-type O-glycans.

**Glycans in mature NPY unfolds its amphipathic alpha-helix.** GalNAc-type O-glycans are most often found in unstructured regions of glycoproteins[15,40]. Surprisingly, the conserved glycans in all the members of the GCG, NPY, and calcitonin families are positioned at the C- or N-terminus of amphipathic alpha-helices (Supplementary Fig. 7a–c). The alpha-helix is important for receptor recognition and activation[41], and specifically for NPY, the alpha-helix is suggested to play a crucial role in prompting a two-step receptor-binding mechanism. Here, NPY first binds to the lipid membrane to increase its effective concentration[41,42], followed by a lateral diffusion on the membrane to the receptor, where NPY binds its receptors and activates intracellular signaling cascades. To explore the potential structural impact of O-glycans on NPY, we measured mature NPY in solution without and with O-glycans at Thr32 using circular dichroism (CD) spectroscopy. NPY showed 33% alpha-helical content in line with

previous studies[43], whereas NPY^Thr32 glycovariants with O-glycans in the alpha-helical interface decreased the alpha-helical content to 23–26%, seen as an increase in ellipticity at 222 nm (Fig. 5c, Supplementary Fig. 7d). Interestingly, where the Tn structure produced a major change, elongation to T and ST structures did not further change the secondary structure dramatically, which is in line with previous studies of mucin O-glycopeptides[44].

**O-glycans modulate peptide hormone stability in vitro.** Most peptide hormones are rapidly degraded with half-lives of only a few minutes[11,13]. O-glycans in close proximity to proteolytic processing sites can modulate the rate of processing[20]. To study the effects of O-glycans on peptide hormones, we subjected selected glycosylated peptide hormones to in vitro degradation by neprilysin (NEP) or dipeptidyl peptidase IV (DPP-IV) enzymes, which are known to metabolize a large number of peptide hormones and other bioactive peptides in vivo, including PYY, VIP, GLP-1, SCT, and GAL[45,46]. Both DPP-IV and NEP degraded all tested non-glycosylated peptides fully or partially within 15–30 min, while the corresponding glycopeptides demonstrated increased stability for 60–120 min (Fig. 6a, Supplementary Fig. 8). In some cases, the intact, non-degraded glycopeptide hormones were still detectable even after overnight incubations. The protective effects seemed to be most profound with the larger sialyl-T glycoforms.

**GLP-1^Thr7 exhibits increased circulation time in vivo.** Intact GLP-1 (7–36 amide) is rapidly cleared from circulation with a half-life of ~2 min[47]. To evaluate the effect of O-glycosylation on GLP-1 clearance in vivo, we measured the half-life of GLP-1^Thr7 glycoforms after administration of an i.v. bolus (20 nmol/kg) of GLP-1 to male Wistar rats ($n = 8$; Fig. 6b). In agreement with previous studies, GLP-1 and immunoreactive metabolites were rapidly cleared from circulation with a half-life of $t_{1/2} = 2.7$ min when fitted to a one-phase decay model, while GLP-1^Thr7 glycoforms showed significantly slower elimination with half-lives of $t_{1/2} = 6.4$ min (T-variant) and $t_{1/2} = 22.7$ min (ST variant), respectively. As a result, non-glycosylated GLP-1 was undetectable after the 15 min timepoint from injection, while GLP-1^Thr7-ST remained readily detectable in circulation even after 60 min.

**Discussion**
Here, we developed a strategy to explore O-glycosylation on peptide hormones and demonstrate that almost 33% of all known human peptide hormones may carry O-glycans in four mammalian species investigated. We present a comprehensive atlas of O-glycosites on the large class of peptide hormones, which should find use in wider studies into specific functions of O-glycans on individual hormones. We highlight examples of biological function of site-specific O-glycans in select peptide hormones, demonstrating that O-glycans can modify receptor activation and prevent degradation and clearance. GalNAc-type O-glycosylation is an exceptional form of protein glycosylation in that it is regulated by a large family of up to 20 GalNAc-T isoforms, providing opportunities for a high degree of differential regulation of glycosites on specific proteins in cells[48]. The GalNAc-T isoforms are known to serve co-regulatory roles in fine-tuning functions of specific proteins with clear disease-causing consequences[23,49–51], and we recently demonstrated that graded induction of individual GalNAc-T isoforms in cell models provides corresponding graded O-glycosylation of highly select glycosite substrates specific to the induced GalNAc-T[52]. In the present study, we did not address the biosynthesis of glycosylated peptide hormones and in particular the specific GalNAc-T

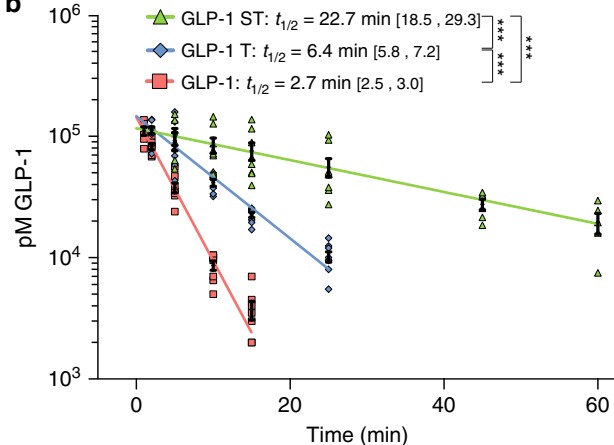

**Fig. 6 Stability of glycosylated peptide hormones in vitro and in vivo. a** Summary of in vitro degradation of peptide hormones using purified NEP or DPP-IV monitored by matrix assisted laser desorption/ionization-time of flight (MALDI-TOF) timecourse assay (See Supplementary Fig. 8 for spectra). Qualitative scoring ++: fully susceptible to degradation, +: reduced susceptibility, −: not susceptible to degradation. **b** Plasma concentrations of GLP-1 or glycoforms over time after intravenous bolus injection (20 nmol per kg) in anesthetized rats ($n = 8$ rats per group). GLP-1 was measured by an assay targeting the amidated C-terminus of the molecule measuring total GLP-1 (intact 7–36 amide and metabolites hereof). Half-lives ($T_{1/2}$) are indicated with 95% confidence intervals. For each glycoform, timepoints represented by $n > 4$ measurements were included in the analysis ($n_{(GLP-1)} = 35$, $n_{(GLP-1\ T)} = 38$, $n_{(GLP-1\ ST)} = 41$ measurements in total). Multiple linear regression was performed in **b**. ns not significant; ***$p < 0.001$ (all indicated p-values < 0.0001). Source data for **b** is provided as a Source data file.

isoforms controlling glycosylation of the identified glycosites. This is a considerable challenge for future studies that should be aided by the presented atlas of glycosites.

Surprisingly, we identified >150 glycosites on mature peptide hormones with many located in the receptor-interacting regions. O-glycans are known to modulate receptor functions, as first described for O-fucosylation of the Notch receptor[53], and more recently for GalNAc-type O-glycosylation of the low-density lipoprotein receptor-related receptors[54–56]. However, to our knowledge, receptors recognizing protein/peptide ligands have not yet been described to exhibit specific recognition of O-glycosylated ligands, apart from the large families of lectin receptors that primarily recognize the glycans[57]. In agreement with this, we found that O-glycans at specific well-conserved amino acids in VIP, GCG, GLP-1, NPY, PYY, and PPY resulted in lower receptor potency in vitro. For GCG and GLP-1, the size of the O-glycan correlated positively with the observed reduction in potency and for VIP all glycoforms diminished activation

similarly (Fig. 4). Among the NPY receptors tested, NPY2R and NPY5R were less affected by O-glycans on NPY and PYY, while O-glycans completely abolished NPY1R and NPY4R activation. Chemical modifications of amino acids in the receptor-binding domain of NPY are known to alter selectivity for the different receptors[58], and our results showing that O-glycans in the ligand-binding region of NPY altered the receptor interaction are well in agreement with this. Although, with the lectin enrichment strategy, we gain the sensitivity needed for glycosite identification, we lose the opportunity for broad assessment of O-glycan stoichiometry. Stoichiometric analysis of O-glycans is one of the major challenges for the field; however, we did demonstrate high glycosylation occupancy for proNPY when expressed in two different cell lines. We expect that future studies will reveal how this reflects the O-glycan occupancy found in circulation or at the local site of secretion.

Interestingly, we observed a substantial overlap between reported phosphorylated peptides from two recent peptidomics studies[9,14], and the identified O-glycosites on peptide hormones (24 out of 91 glycosylated peptide hormones contain both PTMs, see Table 1 and Supplementary Data 5). In several peptide hormones, we found closely positioned phosphosites (superscript P) and O-glycosites (superscript G), including NPY (NLIT^{P/G}RQRY^P and S^GS^GPETLISDLLMRESPT^{G/P}ENVPRT^GRLEDPA), somatostatin (QRS^{P/G}ANS^{P/G}NPA), and vasostatin-2 (VEEPS^PS^GKDVM). This may suggest an interplay between the two PTMs much like the illustrative example of the fibroblast growth factor FGF23[21,59]. Here, O-glycosylation by the GalNAc-T3 isoform inhibits PC processing by furin and adjacent phosphorylation by the Fam20C kinase inhibits glycosylation, and augments processing and inactivation. Thus, glycosylation and phosphorylation may regulate the function of peptide hormones in concert with each other.

Peptide hormones are inherently unstable and circulate in minute amounts. Here, we demonstrated that O-glycans on SCT, VIP, GAL, GLP-1, and PYY protect the peptide hormones from proteolytic degradation in vitro. The most prominent effects were observed with the naturally occurring sialylated ST O-glycan structure, where, e.g., full-length PYY^{Thr32} and GAL^{Ser23} remained readily detectable after 24 h incubation with NEP. In addition, the in vivo circulatory half-life of GLP-1 was increased approximately tenfold compared to the non-glycosylated peptide. These results are well in line with the general concept that protein glycosylation increases stability by shielding the protein backbone from proteolytic cleavage, and also that sialic acid capping of glycans is needed to maintain plasma half-life by, e.g., reducing uptake by the Ashwell–Morell (asialoglycoprotein) receptor[60,61]. Although not addressed here, protein glycosylation can change the cellular uptake and, as shown for an enkephalin-derived glycopeptide, increase both transport across the blood–brain barrier and general bioavailability[62]. Whether naturally occurring O-glycans on peptide hormones affect biodistribution in vivo remains to be elucidated. Our data suggests that O-glycosylation of peptide hormones serves to fine-tune their bioactivities by decreasing receptor stimulation and increasing bioavailability, which may be advantageous for inducing lower and sustainable signals. The O-glycans may also be important for producing selective signals for subsets of related receptors, as we recently described in case of ANP[10]. The NPY family members are ubiquitously expressed in the body, and act as neurotransmitters to regulate a vast array of physiological processes via binding and signaling through the Gi-coupled NPY receptors (NPY1R, NPY2R, NPY4R, or NPY5R), and PTMs may play a role in orchestrating selectivity as observed for the N-terminal truncation by DPP-IV[34]. Another example is GLP-1, whose primary functions include increase of insulin secretion (i.e., to act as an incretin), inhibition of gastrointestinal

motility and physiological regulation of appetite, where PTMs may be involved in directing the response.

The great therapeutic potential of peptide hormones was realized almost a century ago and candidate drug designs continue to emerge[2]. Basing peptide therapeutics on naturally occurring biomolecules is a favorable approach since this often leads to lower toxicity and immunogenicity, as well as higher selectivity, and thus predictable physiological actions and behavior[2]. However, probing the function and efficacy of unmodified peptides has demonstrated low biostability and circulation time, and therefore low efficacy[2,63]. A number of studies have explored the chemical introduction of artificial glycans on peptide hormones[64–67], and interestingly all studies find—very much in line with what we show here for the naturally occurring O-glycans—that the analogs chemically modified with artificial glycan structures are less or equally potent in in vitro receptor activation assays, yet more potent and stable in vivo.

Peptide hormones are frequently detected in clinical immunoassays and the presence of O-glycans on peptide hormones may mask antibody epitopes, which leads to underestimated concentrations as previously demonstrated for NT-proBNP[68,69]. Thus, the atlas may also serve as a guide in the strategic design of biomarker assays.

In summary, we provide a comprehensive atlas of O-glycosites on peptide hormones, and illustrate examples of functions of site-specific O-glycans for processing, bioavailability and bioactivity of select peptides. Our results suggest that O-glycosylation plays much wider roles in regulating and tuning the functions of peptide hormones than currently acknowledged, and considering O-glycans in studies, biomarker assays, and rational drug designs may be fruitful as recently demonstrated for ANP.

## Methods

**Ethical considerations**. Rat in vivo studies and extraction of porcine tissues were conducted with permission from the Danish Animal Experiments Inspectorate (2013-15-2934-00833), and in accordance with the guidelines of Danish legislation governing animal experimentation (1987) and the National Institutes of Health (publication number 85-23). The Regional Committee on Health Research Ethics approved the use of human tissue (KF 01287197), written informed consent was obtained from all study participants, and the study abides by the Declaration of Helsinki principles. Plasma and CSF samples were collected with written informed consent, and subsequently pooled and anonymized. The National Committee on Health Research Ethics has evaluated that the use of these samples for the glycoproteomics study did not need approval because of the anonymization of patient samples.

**Tissue extraction**. Porcine brain, cerebellum, ileum, and pancreas tissue from female LYD pigs of ~30 kg, as well as atrial appendage tissue from newborn piglets were extracted according to standard protocols. Prostate tissue specimens were collected from 25 patients undergoing radical prostatectomy for cancer (median age 65 years (range 54–72))[70]. Aliquots of frozen tissue were crushed using a CryoPrep tissue extractor (Covaris, Woburn, Massachusetts) and proteins were extracted in boiling water for 20 min, followed by homogenization with an Ultra-Turrax homogenizer (IKA, Staufen, Germany; Supplementary Data 2 and 3: LMWE H_2O). After 30 min centrifugation at $13,000 \times g$, supernatants were collected. The ileum sample pellet was further extracted in 0.5 M acetic acid (AcOH) at room temperature (RT) for 30 min (Supplementary Data 2 and 3: LMWE AcOH) followed by centrifugation as above. For acidic ethanol extractions, ileum and pancreatic tissues were homogenized and extracted by rotating at 4 °C for 4 h in 0.18 M HCl/70% ethanol (Supplementary Data 2 and 3: LMWE EtOH/HCl). The samples were adjusted to pH 7 (NaOH), followed by centrifugation as above. A fraction of the brain, cerebellum, prostate gland, and ileum water extract underwent additional low-molecular-weight enrichment by either acetone precipitation (Supplementary Data 2 and 3: LMWE acetone) or acetone precipitation after acidification (Supplementary Data 2 and 3: LMWE acetone AcOH). Precipitation was performed either before or after adjusting to 0.5 M AcOH by addition of ice cold acetone (final v/v 67%), followed by incubation for 1 h at −20 °C, and subsequent centrifugation at $16.000 \times g$ for 30 min (supernatants were collected). All supernatants were lyophilized and reconstituted in Milli-Q water followed by protein concentration determination by Pierce BCA protein assay kit (Pierce, Thermo Fischer Scientific), except the water extracts where lyophilization and reconstitution was omitted.

Rats were sacrificed and pancreas, heart, cerebellum, and brain tissues were dissected free and snap-frozen in liquid nitrogen. Frozen tissues were pulverized

using a mortar and pestle in liquid nitrogen. Proteins were extracted from ~100 mg tissue in 300 μL 50 mM ammonium bicarbonate containing 1% RapiGest SF Surfactant (Waters, from here on: Rapigest), using an IKA Ultra Turbax blender at maximum speed for 20 s followed by 30 s of sonication using a Sonic Dismembrator (Fischer Scientific). Samples were diluted to 0.1% Rapigest in 50 mM ammonium bicarbonate and immediately subjected to enzymatic digestion and desialylation.

**Enzyme digestion and desialylation**. All extracted samples were adjusted to 50 mM ammonium bicarbonate, heated for 10 min at 80 °C, followed by reduction with 5 mM dithiothreitol (60 °C, 30 min) and alkylation with 10 mM iodoaceta-mide (30 min, RT, in the dark). Subsequently, samples were incubated with trypsin, chymotrypsin or GluC (Roche; 37 °C, overnight, 1 μg enzyme per 100 μg protein). The following day, the enzyme reaction was quenched and Rapigest was pre-cipitated by acidifying with trifluoroacetic acid (TFA). The solution was cleared by centrifugation ($10,000 \times g$, 10 min) and peptides were purified on C18 Sep-Pak columns (Waters), and dried down using a SpeedVac vacuum concentrator (Thermo Fischer Scientific). If not already desialylated, the dried peptides were resuspended in 1 mL 50 mM sodium acetate (pH 5.5) containing 0.1 U/mL neur-aminidase (Sigma, N3001) followed by incubation at 37 °C for 1 h, purified by C18 Sep-Pak columns and dried down.

In the case of rat tissues, 200 μg digest was labeled with TMTsixplex or TMT10plex (Thermo Fischer Scientific) according to manufacturer's instructions (Supplementary Data 2 and 3: Rapigest).

**Cell protein extraction**. STC-1 (CRL-3254) cells derived from mouse invasive small intestinal neuroendocrine carcinoma and N2a (CCL-131) cells from mouse neuroblastoma were obtained from ATCC and maintained in Dulbecco's modified Eagle's medium (DMEM) (Sigma) supplemented with 10% fetal bovine serum (Sigma) and 4 mM GlutaMAX (Gibco) at 37 °C and 5% $CO_2$. Conditioned media cleared from dead cells and debris obtained from 2× T175 flasks cultured for 48–72 h were dialyzed, neuraminidase (N3001, *Clostridium perfringens* neur-aminidase type VI, Sigma) treated, and enriched for glycoprotein by capture on short (300 μl beads contained in 1 ml syringes) PNA agarose columns (Vector labs). Glycoproteins were eluted by heating the lectin (2 × 90 °C, 10 min) in 50 mM ammonium bicarbonate containing 0.05 % Rapigest. Total cell lysates were obtained by washing a monolayer of cells in ice cold phosphate-buffered saline (PBS), scraping off the cells and adding 2 ml 0.05–0.1% Rapigest to solubilize the cell pellet. The resulting suspension was sonicated and cleared by centrifugation followed by enzymatic digestion and desialylation as described above.

**Biofluid extraction**. Blood for O-glycoproteomic plasma analyses was collected and pooled from two healthy volunteers into EDTA-coated tubes (K2E K2EDTA Vacuette) followed by centrifugation at $5000 \times g$ for 10 min and stored at −20 °C until use. CSF was obtained and pooled from 50 patients that underwent lumbar puncture. The samples were anonymized before use. Both crude and LMWF-enriched plasma and CSF were subjected to the O-glycoproteome workflow. For LMWF enrichment, 40 mL CSF or a volume of plasma containing ~60 mg of proteins (measured by Pierce BCA protein assay kit) was precipitated by adding two parts of 96% ethanol followed by incubation at RT for 30 min. Samples were centrifuged at $10,000 \times g$ for 10 min, and the supernatant was lyophilized and resuspended in 0.05% Rapigest followed by desialylation in 50 mM sodium acetate buffer with 0.1 U/mL neuraminidase (Sigma, N3001). The LMWF-enriched sam-ples were further enriched for O-glycopeptides on short PNA columns as done for the cell secretomes. In parallel, 5 mg of total protein (as determined by Pierce BCA protein assay kit) from non-LMWF-enriched biofluid samples was desialylated as described for the LMWF-enriched samples, before enzymatic degradation omitting the glycoprotein enrichment on short PNA columns.

**LWAC O-glycopeptide enrichment**. Agarose bound lectins PNA (binds galactosyl (β-1,3) N-acetylgalactosamine, T-antigen), Jacalin (binds galactosyl (β-1,3) N-acetylgalactosamine, T-antigen, or α-N-acetylgalactosamine, Tn-antigen[71]), or VVA (α-N-acetylgalactosamine, Tn-antigen) were obtained from Vector Labora-tories. Dried samples were reconstituted in 2 mL lectin-binding buffer (PNA-binding buffer 10 mM HEPES (pH 7.4), 150 mM NaCl, 0.1 mM $CaCl_2$, and 0.01 mM $MnCl_2$; Jacalin-binding buffer 175 mM Tris (pH 7.5); VVA-binding buffer 20 mM Tris-HCl (pH 7.4), 150 mM NaCl, 1 mM $CaCl_2/MgCl_2/MnCl_2/ZnCl_2$, and 1 M urea), filtered through a 0.45 μm spin column and injected onto a pre-equilibrated 2.6 m long column packed with lectin-bound (PNA, Jacalin, or VVA) agarose beads at a constant flow rate of 0.1 mL/min. For VVA (N2A cells are naturally deficient in O-glycan elongation and produce glycoproteins with truncated Tn (HexNAc) glycans), the column was first washed for 3× column volumes (CV) in 0.4 M glucose and glycopeptides were eluted with 2× CV 0.2 M GalNAc and 1 CV 0.4 M GalNAc. For PNA and Jacalin LWAC, the column was washed 2× CV in lectin-binding buffer, and glycopeptides were eluted with 2× CV 0.5 M galactose and 1× CV 1 M galactose, respectively. A total of 5% of the elution fractions were analyzed by LC–MS/MS to check for sample complexity. If appropriate, the elution fractions were pooled and fractionated using either high pH-fractionation[52] or isoelectric focusing prior to nLC-MS/MS analysis.

**nLC-MS/MS analysis**. LC–MS/MS was performed on a system composed of an EASY-nLC 1000 (Thermo Fisher Scientific) interfaced via a nanoSpray Flex ion source to an LTQ-Orbitrap Velos pro hybrid spectrometer or Orbitrap Fusion Tribrid (Thermo Fisher Scientific), equipped for both higher energy collisional dissociation (HCD) and electron transfer dissociation (ETD) modes, enabling peptide sequence analysis with and without retention of glycan site-specific frag-ments, respectively. The nLC was operated in a one-column setup with an ana-lytical column (20 cm length, 75 μm inner diameter) packed with C18 reverse phase material (1.9-μm particle size, ReproSil-Pur, Dr. Maisch, Ammerbuch Entringen, Germany). Each sample dissolved in 0.1% formic acid was injected onto the column and eluted in a gradient from 2% to 30% B in 105 min (2 h method) or 165 min (3 h method), from 30% to 100% B in 5 min and 100% B for 10 min at 200 nl min$^{-1}$ (solvent A, 100% $H_2O$; solvent B, 100% acetonitrile; both containing 0.1% (v/v) formic acid). A data-dependent mass spectral acquisition routine, ETD triggering of subsequent HCD scan, was used for all runs. Briefly, a precursor MS1 scan ($m/z$ 350–1700) of intact peptides was acquired in the Orbitrap at a resolution setting of 30,000 (Velos Pro) or 120,000 (Fusion), followed by Orbitrap ETD-MS2 at a resolution setting of 15,000 (Velos Pro) or 60,000 (Fusion; $m/z$ of 120–2000) of the five most abundant multiply charged precursors in the MS1 spectrum; this event was followed up by an HCD-MS2 fragmentation at a resolution setting of 7500 (Velos Pro) or 60,000 (Fusion; $m/z$ of 120–2,000) for the same precursor ion. In cases where preliminary screening of fractions for glyco-peptide enrichment was carried out prior to IEF, the ETD-MS2 step was omitted, and HCD-MS2 at a resolution setting of 7500 (Velos Pro) or 60,000 (Fusion; $m/z$ 120–2000) of the ten most abundant multiply charged precursors was acquired (top ten method). These HCD-MS2 spectra were simply screened for the appearance of the HexNAc oxonium fragment ions at $m/z$ 204.086.

**Data analysis**. The raw data were processed using Proteome Discoverer 1.4 software (Thermo Fisher Scientific) and searched against the human, porcine, mouse, or rat-specific Uniprot databases downloaded in January 2013. The Sequest HT search node was used for HCD and ETD data. In all cases, the precursor mass tolerance was set to 15 p.p.m. and fragment ion mass tolerance to 20 millimass units. Carbamidomethy-lation on cysteine residues was used as a fixed modification. Methionine oxidation, and HexNAc or HexHexNAc attachment to serine, threonine, or tyrosine were used as variable modifications. In the case of the VVA-enriched N2a cell line samples only HexNAc (Ser, Thr, and Tyr) and methionine oxidation were used as variable mod-ifications. As an additional preprocessing procedure, all HCD spectra showing the presence of fragment ions at $m/z$ 204.08 were extracted into a single.mgf file, and the exact mass of 1×, 2×, 3×, and 4× HexNAc, Hex(1)HexNAc(1) units or the different combinations of the two modifications were subtracted from the corresponding pre-cursor ion mass, generating a distinct file for each subtracted glycan mass[38]. In this process, a suffix of _#xT and/or _#xTn was added to the filename of the generated.mgf files, indicating the number of subtracted masses of Hex(1)HexNAc(1) and/or Hex-NAc units, respectively. In the case of VVA-enriched samples, the subtraction was only performed for HexNAc, resulting in a _#xGalNAc suffix added to the.mgf file-name. These preprocessed data files were submitted to a Sequest HT node under the conditions mentioned above, without considering HexNAc or Hex(1)HexNAc(1) modifications. Unassigned spectra were submitted to a second Sequest HT node using the same parameters as above with the exception of performing the search using semi-specific trypsin/chymotrypsin/GluC cleavage; all spectra were searched against a decoy database using a target false discovery rate of 1%. The final list was filtered against the human part of the NeuroPep database[24] to include only peptide hormones.

**Multiple sequence alignment**. All alignments were performed in ClustalO using the peptide sequences of *Homo sapiens*, *Sus scrofa*, *Mus musculus*, and *Rattus norvegicus*.

**Immunoblotting**. PreproNPY-plasmid (Human, Uniprot: P01303) was obtained from Genewiz (www.genewiz.com) in the pUC57 framework and subcloned into pcDNA3-plasmid (Invitrogen, Thermo Fischer Scientific) with Bam/NotI restric-tion enzymes (New England Biolabs) followed by ligation with T4 DNA ligase (Thermo Fisher Scientific). STC-1 cells[72] stably transfected with preproNPY (maintained in 700 μg/mL Zeocin (Invitrogen, Sigma)) were seeded out in six-well dishes and grown to 50% confluency. Cells were incubated in Opti-Mem media (Sigma) for 72 h to accumulate secreted proteins.

HEK293-6E (obtained through a license agreement with Dr. Yves Durocher, Bioprocédés Institute de recherche en biotechnologie, Montréal) were grown in suspension in serum-free F17 medium (Invitrogen, Thermo Fischer Scientific) supplemented with 0.1% Kolliphor P188 (Sigma) and 4 mM GlutaMax (Gibco, Sigma) at 37 °C and 5% $CO_2$ under constant agitation (120 r.p.m.). A total of 15 mL of HEK293-6E cells ($1 \times 10^6$ cells/mL) were transfected with preproNPY by adding combined 15 μg DNA and 90 μg of polyethyleneimine-25K (Polysciences, Warrenton, PA, USA), co-incubated at RT for 10 min in 500 μL Opti-Mem. HEK293-6E cells were cultured for 5 days before harvest of the conditioned media.

For enzymatic treatment, the conditioned medium was adjusted to 20 mM Tris (pH 6.8). STC-1 samples were treated with 0.05 mU/μL neuraminidase (Sigma, N3001) overnight (37 °C), and HEK-6E samples were treated with 0.05 mU/μL neuraminidase (Sigma, N3001) and 4 U/μL OpeRATOR for 6 h (37 °C).

Subsequently, 200 μL of each STC-1 sample was concentrated through trichloroacetic acid precipitation (final v/v 10%) followed by centrifugation (16,000 × g, 10 min), and resuspension of the pellet in 15 μL 67 mM Tris (pH 8), 1× NuPage® LDS-buffer (Thermo Fischer Scientific) and 1× Bolt Sample Reducing Agent (Thermo Fischer Scientific) to allow for immunoblotting of all 200 μL/lane. A total of 10 μL of each HEK-6E sample was stopped by addition of 3.5 μL 4× LDS buffer and 1.5 μL 10x Bolt Sample Reducing Agent to a final volume of 15 μL and blotted without upconcentration.

Samples were boiled 10 min at 80 °C, prior to separation on 4–12% Bis-tris gels (NuPage, Thermo Fischer Scientific) and blotted onto nitrocellulose membrane for 11 min at 20 V, using the semi-dry Iblot2 system (Thermo Fischer Scientific). Antibody staining was performed according to the modified protocol by Okita et al.[73]. Membranes were blocked 5 min in 1% milk, 0.1% bovine serum albumin (Sigma; w/v) in Tris-buffered saline with 0.05% Tween-20 (TBS-T) followed by 3 min wash in PBS with 0.05% Tween-20 (PBS-T). Blots were fixed in 0.2% glutaraldehyde (Sigma) in PBS-T for 15 min at RT. After three brief washes in PBS-T, antigen retrieval was performed by microwave treatment (600 W, 10 min) in citrate retrieval buffer (10 mM citric acid, 1 mM EDTA, 0.05% Tween-20, pH 6.0), followed by incubation in quenching buffer for 10 min at RT (200 mM glycine in TBS-T, pH 7.4). Blots were incubated overnight with anti-NPY antibody (Cell Signaling Technologies D7Y5A, Rabbit mAb #11976) diluted 1:1000 in the blocking solution at 4 °C. On the next day, the blots were washed in TBS-T followed by 1 h incubation with secondary HRP-conjugated goat antirabbit Ig's (DAKO, P0448, 0,25 μg/mL). The membranes were washed in TBS-T and developed using SuperSignal West Pico PLUS Chemiluminescent Substrate (Thermo Fischer Scientific). Pictures were taken with a LAS-3000 instrument (GE healthcare)

**Purified glycopeptides**. Peptide hormones were obtained from chemical synthesis (Synpeptides) in both non-glycosylated and T-glycosylated forms, except the VIP and SCT peptides that were only obtained in non-glycosylated forms. NPY and PYY were also obtained in their Tn forms (Synpeptides). To obtain Tn-glycosylated variants of VIP, GCG, GAL, SCT, and GLP-1, non-glycosylated peptides were chemoenzymatically glycosylated with GalNAc-T1[25] (25 mM cacodylic acid sodium, pH 7.4, 10 mM MnCl₂, 0.25% Triton X-100, 2 mM UDP-GalNAc (Sigma), 40 μg/mL enzyme, and 1 mg/mL peptide, 37 °C overnight) and C18 high-performance liquid chromatography (HPLC) purified. Correct site incorporation was confirmed by LC–MS. The Tn-glycosylated peptides obtained by either chemical or chemoenzymatic synthesis were chemoenzymatically elongated to the T-variant using the T-synthase[74] (25 mM cacodylic acid sodium, pH 7.4, 10 mM MnCl₂, 0.25% Triton X-100, 2 mM UDP-Gal (Sigma), 4 pmol/h/μL enzyme, and 1 mg/mL peptide, 37 °C overnight) and HPLC purified. A fraction of all T peptides was chemoenzymatically elongated to the ST form using purified ST3Gal1 (50 mM MES pH 6.5, 2 mM CMP-NeuAc (Sigma), 40 μg/mL enzyme, 1 mg/mL peptide, 37 °C overnight)[75]. The glycoproducts from each reaction were separated from non-glycosylated product on C18 HPLC (Thermo, Ultimate™ 3000, Phenomenex, Jupiter, 5 μm, 300 Å, 250 mm) under isocratic conditions (between 20% and 40% acetonitrile dependent on peptide), lyophilized and resuspended in Milli-Q water. Glycopeptide product quantification was performed using HPLC (UV 210 nm) by comparison with a standard curve of weighed non-glycosylated peptide as standard. GCG^Thr7-T was deglycosylated using O-glycanase (Merck, #324716; 50 mM phosphate buffer pH 6.0, 0.13 mU/mL enzyme, 1 mg/mL GCG^Thr7-T, 37 °C for 2 h) and HPLC purified as described above. Deglycosylated GCG precipitated at assay conditions, and was resolubilized by addition of 0.1% TFA prior to HPLC purification.

**Transfection and tissue cultures for receptor activation**. COS-7 (CRL-1651, ATCC) cells were grown in DMEM 1885 supplemented with 10% fetal bovine serum (Sigma), 2 mM glutamine, 180 units/ml penicillin, and 45 g/ml streptomycin at 10% CO₂ and 37 °C. One day after seeding in culture flasks (seeding density of 6 million cells per T175 flask), the cells were transiently transfected with 40 μg receptor DNA for cAMP determination and a combination of 20 μg receptor DNA and 30 μg of pcDNA 1.1 plasmid encoding a chimeric G protein GαΔ6qi4myr[76] for IP3 determination, using the calcium phosphate precipitation method[77].

**IP3 assay**. One day after transfection, COS-7 cells transiently expressing either NPY1R, 2 R, 4 R, or 5 R (35,000 cells/well) were incubated with myo-[³H]inositol (5 μl/ml, 2 μCi/ml) in 0.1 ml of medium overnight in a 96-well plate. The following day, cells were washed twice in PBS and incubated in 0.1 ml of Hanks' balanced salt solution (Invitrogen, Thermo Fischer Scientific) supplemented with 10 mM LiCl at 37 °C in the presence of various concentrations of NPY or PYY, and their respective glycoforms for 90 min (in duplicate determinations). Assay medium was then removed, and cells were extracted by the addition of 50 μl of 10 mM formic acid to each well, followed by incubation on ice for 30–60 min. The [³H]inositol phosphates in the formic acid cell lysates were thereafter quantified by adding yttrium silicate-poly-D-Lys-coated SPA beads[78]. Briefly, 35 μl of cell extracts were mixed with 80 μl of SPA bead suspension in H₂O (12.5 μg/μl) in a white 96-well plate. Plates were sealed and shaken on table shaker for at least 30 min. SPA beads were allowed to settle and react with the extract for at least 8 h before radioactivity was determined using a Packard Top Count NXT scintillation counter (PerkinElmer Life Sciences).

**cAMP assay**. One day after transfection, the COS-7 cells were seeded at 35,000 cells/well in 96-well plates. 24 h later, cells were washed twice with HEPES-buffered saline (in mmol/L: 20 HEPES, 150 NaCl, 0.75 NaH₂PO₄, pH 7.4) and incubated 30 min at 37 °C in HEPES-buffered saline containing 1 mmol/L iso-butylmethylxanthine (IBMX) phosphodiesterase inhibitor (Sigma-Aldrich, Brøndby, Denmark). Peptide hormones, either non-glycosylated or glycosylated on Thr7, were then added (in duplicate determinations) followed by 30 min incubation at 37 °C. Subsequently, the medium was removed and the cells were treated with the enzyme fragment complementation-based cAMP assay according to manufacturer's instructions (HitHunter cAMP XS+assay, DiscoveRx, Fremont, CA). The cAMP content was measured as luminescence using an EnVision 2104 Multitable Platereader (PerkinElmer) with a cAMP standard curve for validation.

**Enzyme assay**. Recombinant protease assays were performed in a MALDI-TOF timecourse assay. A total of 15 μM peptide hormone underwent enzymatic treatment in vitro at 37 °C in either 50 mM Tris, 0.05% Brij-35, pH 9.0 for NEP assays, or 50 mM Tris, pH 8.0 for DPP-IV assays. The amount of enzyme used was optimized to fully digest the non-glycosylated peptide hormone of interest within 1 h of incubation. The final concentrations used in the assays are as follows; NEP (R&D Systems): 150 pg/μL enzyme for VIP, GAL, and SCT assays. A total of 10 ng/μL for GLP-1 assays, and 20 ng/μL for PYY assays; DPP-IV (R&D Systems): 4 ng/μL for VIP assays, 5 ng/μL for GLP-1 assays, and 10 ng/μL for PYY assays. Product development was evaluated by quenching 1 μL reaction in 40 μL 0.1% TFA after 15 min, 30 min, 60 min, 120 min, and 24 h of reaction time, followed by MALDI-TOF analysis. A 1:1 mixture of GLP-1^Thr5 and GLP-1^Thr7 glycosylated variants was used in this assay due to limited amounts of material.

MALDI-TOF-MS was performed in linear positive mode on a Bruker Autoflex instrument (Bruker Daltonik GmbH, Bremen, Germany) by mixing the quenched aliquots with a saturated solution of α-Cyano-4-hydroxycinnamic acid in ACN/H₂O/TFA (70:30:0.1) at a 1:1 ratio on a target steel plate.

**CD spectroscopy**. CD spectra of 20 μM NPY or NPY glycovariants were obtained in 15 mM sodium phosphate buffer (pH 7) at 25 °C between 200 and 260 nm at a scan rate of 50 nm/min. Each spectrum was obtained as an average of 15 accumulations corrected by subtracting the measurements in 15 mM phosphate buffer alone. Measurements were performed in a Jasco-810 spectropolarimeter equipped with Peltier temperature control, using a quartz cuvette of 0.1 cm path length. Alpha-helical content was calculated by the mean residue molar ellipticity at 222 nm as $[(-[\theta]_{222} + 3000)/(36,000 + 3000)] \times 100$[79].

**Half-life of native and glycosylated GLP-1 isoforms in rats**. Male Wistar rats (≈250 g) were obtained from Janvier (Saint Berthevin Cedex, France) and housed in pairs under standard conditions with ad libitum access to chow and water. Rats were acclimatized for at least 1 week before the study. Studies were carried out on two occasions, matching on each occasion the number of rats in each group, while at the same time avoiding rats from the same cage receiving the same test solution. Rat body weight did not differ between treatment groups (343 ± 14.36 g vs. 333.9 ± 15.74 g vs. 353.3 ± 12.57 g, $p > 0.359$). On the day of study, non-fasted rats were anesthetized by a subcutaneous injection with Hypnorm/Midazolam (0.3 ml per 100 g body weight) and placed on a heated operating table (37 °C). The abdominal cavity was opened by a midline incision and a plastic catheter was inserted in the portal vein. Blood was collected (~300 μl per timepoint) through the catheter into ice cold EDTA-coated tubes that were immediately placed one ice. Samples were collected at timepoint −5, 0, 1, 2, 5, 10, 15, 25, 45, and 60 min. In between samples, ~200 μL PBS were injected to flush the catheter and replace lost fluid. Total blood volume withdrawal was ~3 ml, corresponding to ~15% of the total blood volume (estimated by: blood volume (ml) = 0.06 × body weight (g) + 0.77)[80]. Immediately after the 0 min sample collection, one of three test compounds was injected (20 nmol per kg body weight) into the inferior vena cava through a 26 Gauge needle. Test compounds consisted of synthetic GLP-1 7–36 amide and two glycosylated GLP-1^Thr7-T and GLP-1^Thr7-ST isoforms, prepared in PBS + 1% Human Serum Albumin (Millipore) and injection volume was ~300 μL. Blood was centrifuged (1650 × g, 4 °C, 15 min) within 30 min after collection, and plasma was transferred into clean Eppendorf tubes and stored at −20 °C until hormone analysis. At end of study, rats were sacrificed by diaphragm perforation.

**Biochemical measurements**. Plasma concentrations of total GLP-1 were measured using an in-house assay (code: 89390), employing an antibody diluted 1:200,000 specifically targeting the amidated C-terminus of the molecule, thus measuring both intact GLP-1 (7–36 amide), the primary metabolite (9–36 amide) and other potential N- or mid-terminally truncated isoforms[81]. Non-extracted samples were diluted 500× in assay buffer to stay within the sensitive range of the assay, and to eliminate potential cross-reaction between the antibodies and non-GLP-1 related plasma molecules (matrix effect).

**Statistical analysis**. All parameters and errors were determined with the GraphPad Prism version 8 software, except half-life determinations that were performed with the R Studio software.

Within each independent receptor activation assay, a nonlinear regression (three parameter logistic fit with Hill coefficient = 1) was fitted to the raw data of the non-glycosylated agonist, and top (100% receptor activation) and bottom (0% receptor activation) parameters were used for normalization of all data within that assay. LogEC50's of the normalized values were determined by nonlinear regression (constraining the Hill coefficient = 1 and the top ($E_{max}$) to 100% receptor activation). For $E_{max}$ determinations, the same analysis was repeated without constraint on the top parameter.

Half-life was determined individually for each GLP-1 variant using the linear model command (lm, Concentration ~ Timepoint) on ln-transformed data. Only data points represented by $n > 4$ were included in the analysis. To test the difference in rate constant ($k$), a multiple linear model (lm) was fitted to the pooled data with glycoform set as interaction (lm, Concentration ~ Timepoint * Glycoform). The analysis was repeated while removing the non-glycosylated peptide data to test for difference between the GLP-1 T and ST forms (***$p < 0.001$; **$p < 0.01$; *$p < 0.05$).

When comparing multiple groups, one-way ANOVA was performed with Tukey's post hoc test for testing difference between multiple groups (ns, not significant; ***$p < 0.001$; **$p < 0.01$; *$p < 0.05$ (two tailed)). Comparison of exactly two groups was performed with two-tailed Student's $t$-test (ns, not significant; ***$p < 0.001$; **$p < 0.01$; *$p < 0.05$ (two tailed)).

**Reporting summary**. Further information on research design is available in the Nature Research Reporting Summary linked to this article.

## Data availability

The MS proteomics data underlying Fig. 1, Table 1, Figs. 2 and 3, and Supplementary Figs. 1–3 have been deposited to the ProteomeXchange Consortium via the MassIVE partner repository (massive.ucsd.edu) with the dataset identifier PXD018560. Data are also available at the MassIVE repository with identifier MSV000085289. Source data are provided with this paper. All other data are available in the main text, the Supplementary Materials, or from the corresponding author on reasonable request.

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

## Acknowledgements

This work was supported by the Novo Nordisk Foundation, the Lundbeck Foundation, Danish National Research Foundation Grant DNRF107.

## Author contributions

T.D.M. designed, planned, performed experiments, and wrote the manuscript. L.H.H. and R.E.K. designed and performed experiments. Z.Y. and S.Y.V. planned and performed experiments. S.J., D.B.A., H.J.J., and T.J. performed experiments. J.H., J.P.G., C.M., M.M.R., J.J.H., and C.G. designed experiments. K.T.S. designed, planned, and wrote the manuscript.

## Competing interests

The University of Copenhagen has filed patent applications EP2018/083325 partly on the basis of this work. T.D.M., L.H.H., S.Y.V., J.P.G., C.G., and K.T.S. are named inventors on this application. The other authors declare no competing interests.
