## [Peer Review File · Nature Communications]

Reviewers' comments:

Reviewer #1 (Remarks to the Author):

K. Schjoldager and her colleagues present an impressive study on the identification of O-glycosylated peptide hormones from a wide array of mammalian tissues, body fluids and cell extracts. The authors used several different sample preparation protocols to isolate and fractionate proteolytic digests using ETD and HCD MS2 activation for glycopeptide identification and site assignments.

The authors convincingly demonstrate that O-glycosylation modulates hormone receptor activation and the secondary structure of the peptide hormones. They have also shown that O-glycosylation results in higher in vitro hormone stability, and using GLP-1Thr7 they also demonstrate increased in vivo circulation time. The results provide new insights into the role of O-glycosylation in general and might have high impact on rational drug design and publication in Nature Communications is justified after addressing the reviewers' comments.

My only major issue is that it is very tedious if not impossible to check the reliability of the glycopeptide identifications and STable2 (extended atlas tab) needs to be corrected accordingly. I wanted to check a few identifications and among others, I chose amylin glycopeptide identifications. To do this, I had to calculate precursor m/z, guess for the correct charge state, and look up all potential spectra in the specified raw files (this latter is only included in the ProteomeXchange website along with the raw files). The authors should include the precursor m/z, charge state, and scan number of the identifications along with some measure to estimate the quality of the identifications such as peptide score. It would be also beneficial to include the retention time for further corroboration of the identifications.

Minor issues and questions

1. The title of the manuscript is somewhat misleading: „An atlas of O-glycans on peptide hormones” ... implies that the O-glycan structures were also determined. However, this is not the case – the sample preparation allows for the identification of mucin core-1 type glycopeptides only, and the native glycan structures are also unknown as the sialic acids have been removed during the sample preparation.

2. I found Figure 2 difficult to interpret. The peptide hormone families on Figure 2 are only partly overlapping with hormone family names reported in Table 1 and also the heights of the columns in Figure 2 does not seem to correspond to the number of hormone family numbers listed in Table 1. For example, Table 1 lists 7 glucagon-family hormones but Figure 2 indicates some 10+ family members. It is also not clear how NetOGlyc prediction contributes to the frequency of glycosylation and whether there is overlap between the yellow and gray fields (e.g. NetOGlyc prediction is only considered for peptide hormones not found glycosylated?). Please clarify.

3. The authors chose NPY to demonstrate the high extent of O-glycosylation in the SCT-1 cell line. They show that upon neuraminidase treatment the 2 NPY-related bands collapse into one band on SDS-PAGE and also use Operator to further corroborate their observations. In my opinion, NPY is not an ideal candidate to use Operator for demonstrating glycosylation. Collapsing of the 2 NPY bands into a single one upon neuraminidase treatment indeed indicates that (at least) 2 different glycoforms of NPY were present in the cell secretome and the difference lies in the sialylation status of these glycoforms. I do not see how Operator treatment further corroborates this observation, especially because the gel shift between the Operator treated vs untreated samples is not convincing. If it is

accepted that NPY in the Neu+Operator treated sample runs to a lower molecular mass compared to the Neu-treated sample than it should also be observed that the Neu-treated sample is between the two glycoforms of NPY that does not make sense. As both glycosylation sites are near to the mature peptide's termini, the gel shift induced by Operator treatment is simply too small to convincingly demonstrate the cleavage.

4. Figure S2 illustrates O-glycosylation of selected peptide hormone families and figure legends imply that glycosylation sites identified in the specified sequence or in the paralog hormone are distinguished. This labeling scheme does not seem to be consistent. For example, the NPY sequence provided (YPSKPDNPGEDAPAEDMARYYSALRHYINLITRQRY) is the human (or rat) sequence and it is indicated that S-3 and T-32 were identified O-glycosylated. However, the N-terminal part of this hormone was identified only as YPSKPDNPGEDAPAEDLAR, both glycosylated and unglycosylated, from pig brain. Why is this glycosylation site depicted as yellow square instead of the empty square specified for homologs? Similarly, for the PYY hormone the human sequence is depicted but glycopeptides have been identified only from the pig and mouse samples, while all „amylin“ glycopeptides were identified from pig, rat or mouse. Please clarify.

5. The reference list should be checked. For example, in the results section, ref #24 is cited for the NeuroPep database, the corresponding paper is ref #25.

6. Rather unexpectedly, over 20% of the tissue- or biofluid related glycopeptide PSMs (315/1405) carry only the core HexNAc modification (some up to 3*HexNAc) (see MS peptides guide xls, ProteomeXchange website) and the vast majority of these glycopeptides were isolated using Jacalin. I would not expect this high extent of truncated glycans and Jacalin should not enrich these glycopeptides efficiently. The methods section does not indicate galactosidase treatment either. On the other hand, there is no indication of any HexNAc-only glycopeptide identifications from the VVA-fractionated samples. All this seems to undermine the reliability of the identifications. Please explain. (I tried to figure out sample preparation method from the LC-MS raw file names. It would be very helpful if raw data were linked to the corresponding sample preparation method.)

7. A uniform code should be used for the peptide modifications (see MS peptides guide xls, ProteomeXchange website)— in some peptides the modifications are indicated by lower case characters while in other peptides the modifications are not indicated in the peptide sequence, and the glycan is specified as either Hex(1)HexNAc(1) or Hex1HexNAc1. Finally, there are PSMs in ST2 indicating that part of the data were acquired on TMT-labeled peptides. However, neither experimental details nor results on these experiments were included in the manuscript. There are a

few PSMs with the monosialo core-1 glycan („Hex(1)HexNAc(1)NeuAc(1)”) although only the Tn and T antigens were allowed as glycan modifications according to the methods description.

8. Did the authors try other glycopeptide enrichment methods in order to identify O-glycan structures other than the mucin core-1 type? For example, wheat germ agglutinin binds a wide array of glycan structures albeit with low affinity and interference from N-glycopeptides should be eliminated.

Reviewer #2 (Remarks to the Author):

In this manuscript the authors analyze the presence and function of mucin-type O-glycosylation on peptide hormones. In a paper published earlier this year, the authors had found O-glycosylation on atrial natriuretic peptide that modulated its function, and they wanted to know if other peptide hormones could similarly be modified. Very few peptide hormones have been examined for glycosylation. They used an unbiased approach to identify O-glycosylated peptide hormones by fractionating cells, biofluids, and tissues to enrich low molecular weight proteins (which should contain peptide hormones) and using the well-described targeted glycoproteomics methods they (the Copenhagen group) have previously developed to identify O-glycosylated peptides (removing sialic acids with neuraminidase, digesting with proteases, enriching for O-glycosylated peptides with lectins, and using nano-LC-MS/MS methods to identify glycopeptides). They used the NeuroPep database (database of known neuropeptides and peptide hormones) for bioinformatics. Surprisingly, they showed that nearly a third of the 279 identified peptide hormones are O-glycosylated, and that most of the rest have sequences that are predicted to be modified but were missed in their screens. Many of the glycosylation sites are conserved and occur either in the proprotein processing sites or in the receptor-binding regions. They synthesized several peptide hormones from the Neuropeptide Y and Glucagon families (unmodified or modified with Tn, T or SialylT) and showed that the glycosylation affects proteolytic stability, receptor activation, and/or serum-half life.

These are well done and interesting studies, demonstrating that O-glycosylation of peptide hormones is much more common than previously thought, and that the glycosylation status of these peptides can modulate their biological activities.

Several concerns need to be addressed prior to publication:

1. A major question for biological significance of these modifications is their stoichiometry. Since the methods used to identify glycosites does not provide information regarding stoichiometry, using an alternative method to examine stoichiometry of modification is important. Figure 4f is the one experiment that examines stoichiometry of glycosylation on a single protein: NPY. It is clear that the neuraminidase treatment shifts the upper band to the lower, suggesting that at least half of the protein is glycosylated, but interpretation of the Operator shift is more difficult. The shift is very slight, and since it should be cleaving the peptide N-terminal to the modification site, shouldn't the shift be larger? Could a lectin blot (or some other method) be performed to confirm that both bands are glycosylated? Also, the manuscript would be strengthened by providing data on stoichiometry of an additional peptide hormone (e.g. from the Glucagon family).

2. Supplementary Figure S1a: The figure and legend suggests that up to three different types of protein extraction protocols were used to generated the Low Molecular Weight Extractions (LMWE). The legend refers to Supplementary Table 1 to describe which protocol was used on which sample, but this table is missing. Since the methods are not clear on the three extraction protocols, it would be very helpful to include this table.

3. Third line of Results: The reference for the NeuroPeP database should be 25. The authors should carefully check all of their references.

4. Second paragraph of Results: It is difficult to determine how the numbers in this paragraph are derived from Supplementary Fig. S1b. For instance, 420 glycosites? 34 sites on 16 human prohormones? 386 glycosites on non-human peptide hormones? 350 of 386 conserved?

5. Table 1 needs a legend defining what "X" and "-" mean.

6. Figure 1b: It would help to put gridlines in to make it easier to follow across the rows and columns. Also, the significance of the grayscale is not clear. Does black mean glycopeptides were found in all three extraction methods?

7. Figure 1c: It is hard to tell the Pig and Rat bars apart. Use different colors or patterns?

8. Top of page 6: It would be helpful to add a Supplementary Figure like Supplementary Fig S2 for the POMC, pro-NPY, and somatostatin peptides. It would be easier to see the location of all the Ser/Thr residues in these peptides.

9. Supplementary Figure S3: It would help to have the keys in all of these panels in the same order: unmodified, Tn, T, ST.

10. Page 7:

a. Refer to Figure 4e and Supplementary Figure S3 after this sentence: The NPY family peptide hormones activate the same members of the Gi coupled NPY receptor family (Y1, Y2, Y4, and Y5).

b. Refer to Supplementary Figures S3 and S4 at the end of this sentence: whereas NPYR1 and NPYR4 showed only slight activation at the highest dose of the glycopeptide hormones tested (Supplementary Fig. 4).

c. Refer to panels d and e at the end of this sentence: In general, O-glycans located in the receptor binding domain of the NPY-family members had a bigger impact on receptor activation than glycans in the receptor binding domain of glucagon family members (Fig. 4d).

11. Figure 4h legend: Should reference Supplementary Figure S6 for spectra.

Reviewer #3 (Remarks to the Author):

Madsen et al. applied their expertise in the identification of O-glycosylated products to peptide hormones. The manuscript summarizes a large body of work in which many questions that are of interest to a wide audience were asked in a logical manner and, for the most part, clearly answered. The authors see this study as an “atlas”, or resource for anyone interested in peptide hormones. As such, it needs to be accessible to readers who are not experts in the analysis of O-glycosylation. Without major revisions to the text and figures, the impact that this study should have on the field will be severely compromised. No further experiments are needed. The suggestion that O-glycosylation plays a role in regulating and tuning peptide hormone function is an important one, as is the authors’ suggestion that O-glycosylation sites serve as useful sites for designing more effective peptide based therapeutics.

The amount of work summarized in this report is stunning. The authors’ decision to explore O-glycosylation in multiple species and in multiple tissues greatly strengthens its impact. That said, this is hopefully the start of building an atlas of O-glycosites on peptide hormones; with additional

species and tissues to be explored, there is clearly much more to be learned. The authors establish that many human, pig and rodent peptide hormones and their precursors are O-glycosylated and that this modification is often physiologically significant. The extent to which the sites are conserved amongst family members and across species is stunning, as is localization of the sites to different regions of the precursors.

The chemoenzymatic synthesis of three glycoforms of many of the peptide hormones studied added a great deal of strength to the study. Selecting glucagon family members and NPY family members for detailed study allowed the authors to test experimentally many of the implications of their study. The effect of O-glycosylation on propeptide structure was explored using proNPY and its glycovariants, allowing the authors to assign the effect to the initial Tn structure. Neprilysin and dipeptidyl peptidase IV were used to address the issue of peptide stability in vitro. In vivo half-life experiments with GLP-1 extended the study to more physiologically relevant conditions.

General Comments:

1. The authors state that they demonstrated that almost 33% of all known peptide hormones may carry O-glycans. This is an over-statement. A firm conclusion can be stated for the four species examined and for the limited number of tissues/fluids examined. The suggestion/hypothesis/speculation can then be made that similar results will be observed in the many other organisms/tissues and peptide families that have not yet been examined. The authors state (p.5) that peptide hormones are classified into 46 distinct families, without making it clear that their statement applies only to the closely related species selected for study, not to all eukaryotic peptide hormones. NeuroPeP, which includes over 400 organisms, lists well more than 46 distinct families.

2. To make this body of work accessible to a wider audience, a better introduction to O-glycosylation in general and GalNAc-type O-glycosylation is needed. The symbols used to depict O-glycans (yellow square, yellow circle, etc.) in Fig.3 are not defined until Fig.4b, when the authors finally illustrate the structures about which they have been writing. It is never made clear whether GalNAc-type O-glycosylation accounts for a large or small fraction of O-glycosylation.

3. The authors deal with technical issues about the best ways to identify O-glycosylation sites in peptide hormones and with the results of their analyses. As currently written, these two topics are intermingled - discussion of technical considerations (LMWE strategy vs Rapigest) appears after discussion of conservation across species and location of the sites within propeptides. The manuscript would be much easier to follow if these issues were presented separately. A rationale should be provided for the organisms/tissues/biofluids selected for study. Fig.1 deals with some of the technical issues, but the conclusions are not clearly stated. For example, it is not readily apparent what the 0, 1, 2 and 3 extraction methods referred to are? In Fig.1d, can the authors

suggest why 10 glycosylated peptides were missed by their new scheme, but detected using Rapigest or a cell line?

4. In several instances, the significance of the claims made is not supported by an appropriate statistical analysis. For example, in Fig.4i, error estimates are needed for the half-lives reported, along with statistical tests to establish which differences are significant. Fig.4h reports susceptibility to proteolytic degradation – it is scored as ++ vs + vs -, with no quantitative statement of how + is distinguished from ++. On p.7, the authors state that “increased EC50-values correlated positively with the size of the O-glycan” – the statistical significance of this statement is not clear.

Questions:

1. Methods: lectin beads were typically eluted with the appropriate sugar and then with that same sugar at double the concentration – it is not clear whether the eluates were pooled or analyzed separately?

2. p.17: STC-1 cells were stably transfected with preproNPY. A vector encoding preproNPY was used to transfect the cells – what vector was used? what species of proNPY did it encode? what was the source of the vector?

3. The sources of many materials are not provided.

4. When the NeuroPeP website is mentioned, reference 24 is cited; the actual website should be provided.

5. The reader is never told anything about the two cell lines included in the study (N2A and STC-1) – What is their derivation? How were they obtained? Why were they used?

6. In Fig.4f, the authors use a STC-1 cell line expressing proNPY to explore the extent to which proNPY is O-glycosylated. Two bands are clearly visible on the Western blot and collapse into one after neuraminidase treatment; no explanation is given for the location of this band at a position intermediate to the two bands in the untreated sample. Treatment with neuraminidase followed by OpeRATOR O-glycoprotease produces a slight downward shift in mobility, from which the authors conclude that there are two distinct glycoforms. These experiments are presented without sufficient interpretation/explanation. Not every reader will know what OpeRATOR does. It is impossible to interpret the size shift following OpeRATOR cleavage without guidance from the authors about the expected size shift and the specificity of the NPY antibody being used – would its ability to detect proNPY be affected by OpeRATOR cleavage? Is the reader to conclude that all of the proNPY is O-glycosylated?

7. On p.9, the authors state that “Both DPP-IV and NEP degraded all of the tested non-glycosylated peptides fully or partially within 15-30 min ...”. Based on the Methods section (p.19), the enzyme concentrations were adjusted to achieve this goal, with very different amounts of each protease used for different propeptides. Although it is clear that the glycopeptides are more stable, the

conclusion that the protective effects of the O-glycans increases with the size of the O-glycan structure requires statistical analysis.

Minor points: there are many typographical errors throughout the text

p.5 – third line from bottom – the text ending with reference 27 is not a sentence.

p.9 – 2nd line before the Discussion – “As a results”, instead of “As a result”

P.13: 0.5 M CH₃COOH were added icecold – presumably “were added to ice cold”?

Throughout – abbreviations should be defined at first use and their use minimized (e.g., Rapigest, LWAC, MQ)

p.14: line 2, “as done for the the LMWF-enriched”

p.14: line 3, “icecold PBS, scrabing off” – presumably “ice cold PBS, scraping off”?

p.17 – lines 2&3 of “Purified glycopeptides.” section – “the VIP and secretin peptides which was only” – verb should be plural, “were”. Later in same paragraph, “The Tn-glycosylated peptides was further” – “were”

p.18: “6 mio cells/flask”??

If the “structures” shown in Fig.4a and Fig.4c are not based on actual data for these ligand/receptor interactions, it is not clear that adding a cartoon of the sugar chain increases understanding.

The NetOGlyc algorithm is mentioned, with citation of reference 15; the website used should be indicated.

Fig. S1 – PNA, JAC, VVA lectins – specificity should be stated

Point-by-point response & action list to reviewer's comments

Reviewer #1:

Query 1: My only major issue is that it is very tedious if not impossible to check the reliability of the glycopeptide identifications and STable2 (extended atlas tab) needs to be corrected accordingly. I wanted to check a few identifications and among others, I chose amylin glycopeptide identifications. To do this, I had to calculate precursor m/z, guess for the correct charge state, and look up all potential spectra in the specified raw files (this latter is only included in the ProteomeXchange website along with the raw files). The authors should include the precursor m/z, charge state, and scan number of the identifications along with some measure to estimate the quality of the identifications such as peptide score. It would be also beneficial to include the retention time for further corroboration of the identifications.

Response 1: We agree, thank you.

Action 1: We now include modified Sppl. Data 2 "Extended atlas" and MS peptides guide. We have included precursor m/z, charge state, XCorr-score, Retention time, and scan number in these Tables. In addition, we have uploaded .msf processing files from Proteome Discoverer program into the public repository. All assigned MS/MS can be accessed directly by using the supplied Table guide.

Minor issues and questions

Query 2: The title of the manuscript is somewhat misleading: „An atlas of O-glycans on peptide hormones"... implies that the O-glycan structures were also determined. However, this is not the case – the sample preparation allows for the identification of mucin core-1 type glycopeptides only, and the native glycan structures are also unknown as the sialic acids have been removed during the sample preparation.

Response 2: We agree.

Action 2: The title was changed to "An atlas of O-linked Glycosylation on peptide hormones reveal wide biological roles"

Query 3: I found Figure 2 difficult to interpret. The peptide hormone families on Figure 2 are only partly overlapping with hormone family names reported in Table 1 and also the heights of the columns in Figure 2 does not seem to correspond to the number of hormone family numbers listed in Table 1. For example, Table 1 lists 7 glucagon-family hormones but Figure 2 indicates some 10+ family members. It is also not clear how NetOGlyc prediction contributes to the frequency of glycosylation and whether there is overlap between the yellow and gray fields (e.g. NetOGlyc prediction is only considered for peptide hormones not found glycosylated?). Please clarify.

Response 3: We agree.

Action 3: We have modified Figure 2 legend to clarify that it illustrates the glycosylated part of all the 46 peptide hormone families described in the NeuroPep database, whereas Table 1 lists the glycosylated peptide hormone part only. Furthermore, we have made changes to the figure to illustrate which proportion of the identified sites are also *predicted* by the NetOGlyc 4.0.

Also, to clarify which peptide hormones are *predicted* to carry O-glycans, in Sppl. Data 1 we have included a column that specifies the exact sites predicted by NetOGlyc.

Query 4: The authors chose NPY to demonstrate the high extent of O-glycosylation in the SCT-1 cell line. They show that upon neuraminidase treatment the 2 NPY-related bands collapse into one band on SDS-

PAGE and also use Operator to further corroborate their observations. In my opinion, NPY is not an ideal candidate to use Operator for demonstrating glycosylation. Collapsing of the 2 NPY bands into a single one upon neuraminidase treatment indeed indicates that (at least) 2 different glycoforms of NPY were present in the cell secretome and the difference lies in the sialylation status of these glycoforms. I do not see how Operator treatment further corroborates this observation, especially because the gel shift between the Operator treated vs untreated samples is not convincing. If it is accepted that NPY in the Neu+Operator treated sample runs to a lower molecular mass compared to the Neu-treated sample than it should also be observed that the Neu-treated sample is between the two glycoforms of NPY that does not make sense. As both glycosylation sites are near to the mature peptide's termini, the gel shift induced by Operator treatment is simply too small to convincingly demonstrate the cleavage.

Response 4: We agree. The Operator treatment results are challenging to interpret. In order to visualize NPY by Western blot (Fig. 4f), it was necessary to concentrate the secreted proteins from the cell culture media by TCA precipitation prior to separation by SDS-PAGE. The Operator treatment induced a minor shift in size but most prominently a reduction in band intensity suggesting that some peptide was lost from detection. Now we repeated this experiment with a similar result, but obtained better separation of the non-glycosylated and glycosylated peptide using a different gel-type (4-12% Bis/Tris). However, we were still not able to detect any cleavage products from the Operator reaction. We suspect that the TCA precipitation complicates the detection of the proteolytic fragment. To further explore the Operator cleavage, we expressed proNPY in a high producer cell line HEK293 6E that secreted proNPY in amounts detectable by western blotting without any sample pre-enrichment. The results are similar to what we see in STC-1 cells with one important exception. When treating with Operator the upper band loses intensity and a band of approximately 4-5 KDa appears. We repeated this experiment three times with similar results. By LC-MS/MS we identified O-glycans on both the C-terminus (CPON, Ser^{40/41}, Thr⁵⁵ and Thr⁶¹) and N-terminus (mature, Ser³ and Thr³²) of proNPY of which only Thr³² is conserved among the three family members (NPY, PYY, PP). Both glycosylation of Thr32 and Ser40 or 41 would result in a fragment of approximately 4-5 KDa. Thus, it is not possible to determine the specific cleavage site based on this data, however we can confirm that NPY is GalNAc-type O-glycosylated to approximately 50% in two different cell lines, STC1 and HEK293 6E.

Action 4: We have replaced Fig.4f (below) with a higher resolution blot.

We have added Western blot of HEK293 6E produced proNPY (below) as Supplementary Figure S6a including a schematic drawing S6b visualizing the possible cleavage fragments from the Operator reaction

and corresponding sizes.

Supplementary Figure 6. Western blot showing OperATOR cleaved fragments of glycosylated proNPY. a) Media from HEK293 6E cells transiently transfected with proNPY was either desialylated with neuraminidase (neu), treated with the glycoprotease Operator, or both in combination. b) Schematic illustrating theoretical masses of the N-terminal fragments of proNPY after cleavage by the OperATOR at the N-terminal site of O-glycosites identified in this study. The theoretical mass-change by cleavage at T-glycosylated Ser3 will show approximately same mass as full length non-glycosylated proNPY (+0.1kDa) due to the size of the glycan on the remaining antibody-recognized fragment. Source data for panel a) is provided as a Source Data file.

The Materials and Methods section for immunoblotting has been updated/revised accordingly.

Query 5: Figure S2 illustrates O-glycosylation of selected peptide hormone families and figure legends imply that glycosylation sites identified in the specified sequence or in the paralog hormone are distinguished. This labeling scheme does not seem to be consistent. For example, the NPY sequence provided (YPSKPDNPGEDAPAEDMARYYSALRHYINLITRQRY) is the human (or rat) sequence and it is indicated that S-3 and T-32 were identified O-glycosylated. However, the N-terminal part of this hormone was identified only as YPSKPDNPGEDAPAEDLAR, both glycosylated and unglycosylated, from pig brain. Why is this glycosylation site depicted as yellow square instead of the empty square specified for homologs? Similarly, for the PYY hormone the human sequence is depicted but glycopeptides have been identified only from the pig and mouse samples, while all „amylin” glycopeptides were identified from pig, rat or mouse. Please clarify.

Response 5: This appear to rely on a misinterpretation and we have now modified the text to clarify. We do not report any “unglycosylated” peptides in our dataset, however in supplementary data 2 “Extended atlas” the ambiguously assigned glycosites were left blank in the “modification” column. We speculate that these might have been interpreted as “unglycosylated” peptides.

Action 5: Figure S2 and Legend has been adjusted to clarify that yellow squares represent glycosylated residues identified in peptide hormones in either of the four investigated species and that empty squares are predicted sites based on conservation between paralogs within the family. Furthermore, we have added symbols to clarify which sites have been identified in previous studies (crossed yellow square) and which sites are not conserved in human sequences (faded yellow square).

To clarify the extent (how many sites) and nature (ex. HexHexNAc) of glycosylation found on ambiguously assigned glycopeptides, a column with this information has been added to the extended atlas (suppl. Data 2, column D).

Query 6: The reference list should be checked. For example, in the results section, ref #24 is cited for the NeuroPep database, the corresponding paper is ref #25.

Response 6: Thank you!

Action 6: The Neuropep database is now correctly cited, and all references have now been double checked.

Query 7. Rather unexpectedly, over 20% of the tissue- or biofluid related glycopeptide PSMs (315/1405) carry only the core HexNAc modification (some up to 3*HexNAc) (see MS peptides guide xls, ProteomeXchange website) and the vast majority of these glycopeptides were isolated using Jacalin. I would not expect this high extent of truncated glycans and Jacalin should not enrich these glycopeptides efficiently. The methods section does not indicate galactosidase treatment either. On the other hand, there is no indication of any HexNAc-only glycopeptide identifications from the VVA-fractionated samples. All this seems to undermine the reliability of the identifications.

Please explain. (I tried to figure out sample preparation method from the LC-MS raw file names. It would be very helpful if raw data were linked to the corresponding sample preparation method.)

Response 7: In the study we included N2A cells that are naturally deficient in O-glycan elongation and therefore produce truncated Tn glycoproteins, and a part of the O-glycosites identified with HexNAc alone originates from these cells. Moreover, the binding specificity of Jacalin is high for both GalNAc and Gal-GalNAc (Tachibana et al 2006 (<https://academic.oup.com/glycob/article/16/1/46/651394>), and we have in a number of studies reported efficient enrichment of both glycoforms (King et al., Khetarpal et al., Hintze et al., Hansen et al)). These points were unfortunately not sufficiently clear in the text. Finally, truncated (HexNAc) glycopeptides in total cell lysates are routinely found (Ye et al., Nat Meth 2019, King et al., Blood Adv 2017, Khetarpal et al., Cell Metab 2016), and predicted to represent biosynthetic intermediates in normal cells.

Action 7: To clarify which sample were enriched with which lectin, a column indicating which lectin strategy was used has been implemented in Supplementary data 2 “extended atlas”, column M, and an accompanying text has been included in the main text as well as a reference to the mentioned paper p. 15

“For VVA (N2A cells are naturally deficient in O-glycan elongation and produce glycoproteins with truncated Tn (HexNAc) glycans) the column was first washed for 3x column volume” and

“Agarose bound lectins PNA (binds galactosyl (β -1,3) *N*-acetylgalactosamine, T-antigen), Jacalin (binds galactosyl (β -1,3) *N*-acetylgalactosamine, T-antigen or α -*N*-acetylgalactosamine, Tn-antigen (Tachibana et al., 2006) or VVA (α -*N*-acetylgalactosamine, Tn-antigen) were obtained from Vector Laboratories”

Query 8: A uniform code should be used for the peptide modifications (see MS peptides guide xls, ProteomeXchange website)– in some peptides the modifications are indicated by lower case characters while in other peptides the modifications are not indicated in the peptide sequence, and the glycan is specified as either Hex(1)HexNAc(1) or Hex1HexNAc1. Finally, there are PSMs in ST2 indicating that part of the data was acquired on TMT-labeled peptides. However, neither experimental details nor results on these experiments were included in the manuscript. There are a few PSMs with the monosialo core-1 glycan („Hex(1)HexNAc(1)NeuAc(1)”) although only the Tn and T antigens were allowed as glycan modifications according to the methods description.

Response 8: Thank you, we have now introduced consistent codes.

In a separate study we performed extensive differential quantitative proteomics which is being analyzed and to be summarized later. In the dataset we found 5 unique peptides and for completion we included these in the glycopeptide hormone atlas. The peptides were not quantitatively affected in the experiment; however, the TMT labelling is known to affect ionization and may have improved the identification. We

inadvertently neglected to describe the TMT labeling. For now, we have included these peptides in our analysis, however if the reviewers prefer we can take them out of the collected dataset without major implications on the driving point of our work.

In a few samples the raw data processing also included HexHexNAcNeuAc as a variable modification. This information was also inadvertently omitted.

Action 8: The supplementary data 2 “Extended atlas” as well as MS peptides guide has an updated uniform nomenclature on the glycan structures.

The full dataset is uploaded (also please see Query and Action 1) and available for your perusal and the following sentence has been added to the Materials and Methods section:

“Rats were sacrificed and pancreas, heart, cerebellum and brain tissues were dissected free and snap-frozen in liquid nitrogen. Frozen tissues were pulverized using a mortar and pestle in liquid nitrogen. Rapigest extraction and protein digestion was performed as previously described³¹ and 200 ug digest was labelled with TMTsixplex or TMT10plex according to manufacturer's instructions”.

We have modified the Material and Methods section text to include the description of NeuAc:

“Methionine oxidation, C-terminal amidation (only CSF) and HexNAc or HexHexNAc attachment to serine, threonine or tyrosine were used as variable modifications. In CSF and heart samples also HexHexNAcNeuAc was also considered as a variable modification. As an additional preprocessing procedure, all HCD spectra showing the presence of fragment ions at m/z 204.08 were extracted into a single .mgf file, and the exact mass of 1×, 2×, 3× and 4× HexNAc, HexHexNAc or HexHexNAcNeuAc units was subtracted from the corresponding precursor ion mass, generating four distinct files.”

Query 9. Did the authors try other glycopeptide enrichment methods in order to identify O-glycan structures other than the mucin core-1 type? For example, wheat germ agglutinin binds a wide array of glycan structures albeit with low affinity and interference from N-glycopeptides should be eliminated.

Response 9: Thank you for the interesting suggestion. We agree to the possibility that other O-type glycans may be present on mammalian peptide hormones, however our study focused on the mucin core-1 glycan structures based on the preliminary findings e.g. in Hansen *et al.*, 2019. Thus, expanding the O-glycopeptide hormone atlas with additional types of lectin enrichments has been out of scope for the current manuscript, but is certainly relevant for future studies.

Action 9: None taken

Reviewer #2:

Query 1: A major question for biological significance of these modifications is their stoichiometry. Since the methods used to identify glycosites does not provide information regarding stoichiometry, using an alternative method to examine stoichiometry of modification is important. Figure 4f is the one experiment that examines stoichiometry of glycosylation on a single protein: NPY. It is clear that the neuraminidase treatment shifts the upper band to the lower, suggesting that at least half of the protein is glycosylated, but interpretation of the Operator shift is more difficult. The shift is very slight, and since it should be cleaving the peptide N-terminal to the modification site, shouldn't the shift be larger? Could a lectin blot (or some other method) be performed to confirm that both bands are glycosylated? Also, the manuscript would be strengthened by providing data on stoichiometry of an additional peptide hormone (e.g. from the Glucagon family).

Response 1: We obviously can agree that the stoichiometry of O-glycosylation is an important analytical challenge for the field, however this is still an insurmountable intrinsic challenge because of the need for

enrichment to even detect O-glycopeptides, and not alone for a class of peptide hormones known to be expressed at such low levels that these are often missed in proteomics studies. We spent huge efforts to demonstrate stoichiometry for one example, and while we agree that more examples would be preferable, it is a major task to add additional examples. Nonetheless, we did attempt to demonstrate stoichiometry for additional peptides (VIP, GLP-1 and Secretin), but with the antibodies available to us and use of our expression system and reagents we were unable to visualize these peptides.

The current dataset represents state-of-the-art in the field, and we believe that lectin analyses is not going to aid in any level of quantification beyond what we already demonstrate with neuraminidase treatment. We believe that the value of the results is the discovery of O-glycosylation as a major PTM of neuropeptides, and this serves as a resource for further dissection and analysis of the impact for additional specific examples in the future. We also need to keep in mind that stoichiometry of a given glycosylation site may change in time and space. Peptide hormones are released from cells into blood or interstitial fluid in response to appropriate stimulus where they can assert their biological function. To our knowledge, broad identification of peptide hormones or neuropeptides, let alone glyconeuropeptides, in plasma/serum by mass spectrometry has not yet been achieved, and indeed the majority of the findings presented here originates from cell or tissue lysates. We did include plasma in our analysis which resulted in the identification of a few high abundant (nanomolar) plasma neuropeptides (ex preptin), but none of the peptide hormones in the low picomolar range. Intracellularly both immature biosynthetic intermediates and mature peptides are present which complicates the determination of stoichiometry. To determine biological significant/relevant stoichiometry we will need tools/methods to determine the in vivo/physiological stoichiometry of peptides systemically, or at least secreted at local sites within tissues. We believe this is currently beyond state-of-the art, but we certainly acknowledge the relevance of discussing this aspect and have included the following paragraph to the discussion.

As referred above we do agree that the Operator treatment of proNPY in Figure 4f is challenging to interpret and we have clarified and extended the analysis as described in response to Reviewer 1 Query 4.

Action 1: The following paragraph was added to the discussion:

“Although with the lectin enrichment strategy we gain the sensitivity needed for site identification we lose the opportunity for broad assessment of O-glycan stoichiometry. Stoichiometric analysis of O-glycans is one of the major challenges for the field, however, we did demonstrate high occupancy glycosylation for proNPY when expressed in two different cell lines. How this reflects the O-glycan occupancy found in circulation or at the local site of secretion we expect future studies will reveal.”

Query 2: Supplementary Figure S1a: The figure and legend suggest that up to three different types of protein extraction protocols were used to generate the Low Molecular Weight Extractions (LMWE). The legend refers to Supplementary Table 1 to describe which protocol was used on which sample, but this table is missing. Since the methods are not clear on the three extraction protocols, it would be very helpful to include this table.

Response 2: We agree. This information is presented in supplementary data tab 2 “extended atlas”, but a reference in the text is needed to guide the reader.

Action 2: The legend of Figure S1a now refers to “Sppl. Data 2, extended atlas”.

“a) Schematic workflow for analysis of O-glycosylated peptide hormones. Proteins from biofluids (cerebrospinal fluid, plasma), neuroendocrine tissue (brain, pancreas, ileum, heart, prostate, cerebellum) and neuroendocrine cells (STC-1, N2a) were extracted using up to three different extraction procedures per sample (see Sppl. Data 2, “extended atlas”).”

Query 3: Third line of Results: The reference for the NeuroPep database should be 25. The authors should carefully check all of their references.

Response 3: Thank you.

Action 3: Fixed. We have double checked all references.

Query 4: Second paragraph of Results: It is difficult to determine how the numbers in this paragraph are derived from Supplementary Fig. S1b. For instance, 420 glycosites? 34 sites on 16 human prohormones? 386 glycosites on non-human peptide hormones? 350 of 386 conserved?

Response 4: We agree.

Action 4: In order to clarify quantitative description of glycosites identified we have modified the result section text as follows:

“In total 420 O-glycosites were located on 118 distinct prohormone proteins²⁵. Of these 34 sites were identified on 16 human prohormone sequences (Supplementary Fig. 1b) and 386 glycosites were identified on 102 non-human propeptide hormones. The latter were subjected to prediction of conservation in the human orthologs by sequence alignment analysis revealing that 350 out of 386 were predicted to be conserved in human based on conservative preservation of Ser/Thr residues and adjacent (+/-5) amino acid residues²⁶. In total 384 (34+350) conserved glycosylation sites were included for further analysis (Fig. 1a).”

Query 5: Table 1 needs a legend defining what “X” and “-” mean.

Response 5: We agree.

Action 5: We have included the following sentence in the legend for Table 1 to clarify what “X” and “-” means: “X: Glycopeptide identified in the given tissue; -: no glycopeptide was identified in the given tissue. *Phosphorylated peptides previously identified^{9,14} (see Supplementary Data 4).”

Query 6: Figure 1b: It would help to put gridlines in to make it easier to follow across the rows and columns. Also, the significance of the grayscale is not clear. Does black mean glycopeptides were found in all three extraction methods?

Response 6: We agree.

Action 6: Gridlines has been added to Fig. 1b. The following sentence “b) Heatmap illustrating the identification of glycans on mature peptide hormones across different biofluids, tissues and cell lines. Greyscale intensity illustrates the number of extraction-methods that have resulted in the identification of a given glycopeptide” has been added to the figure legend to clarify the significance of the greyscale.

Query 7: Figure 1c: It is hard to tell the Pig and Rat bars apart. Use different colors or patterns?

Response 7: We agree.

Action 7: We have changed Fig.1c by adding colors which are more easily distinguished from each other.

Query 8: Top of page 6: It would be helpful to add a Supplementary Figure like Supplementary Fig S2 for the POMC, pro-NPY, and somatostatin peptides. It would be easier to see the location of all the Ser/Thr residues in these peptides.

Response 8: We agree.

Action 8: An additional Suppl. Fig. 2 has been produced and added showing the full pro-sequences for POMC, pro-NPY and somatostatin.

Query 9: Supplementary Figure S3: It would help to have the keys in all of these panels in the same order: unmodified, Tn, T, ST.

Response 9: We agree.

Action 9: In all panels in Fig. S3 the order of molecules assayed has been changed into Non-glyc., Tn, T and ST

Query 10: Page 7:

- a. Refer to Figure 4e and Supplementary Figure S3 after this sentence: The NPY family peptide hormones activate the same members of the Gi coupled NPY receptor family (Y1, Y2, Y4, and Y5).
- b. Refer to Supplementary Figures S3 and S4 at the end of this sentence: whereas NPYR1 and NPYR4 showed only slight activation at the highest dose of the glycopeptide hormones tested (Supplementary Fig. 4).
- c. Refer to panels d and e at the end of this sentence: In general, O-glycans located in the receptor binding domain of the NPY-family members had a bigger impact on receptor activation than glycans in the receptor binding domain of glucagon family members (Fig. 4d).

Response 10: Yes, agree.

Action10: Fixed, thank you.

Query 11: Figure 4h legend: Should reference Supplementary Figure S6 for spectra.

Response 11: Agree.

Action11: Fixed, thank you.

Reviewer #3

Query 1: The authors state that they demonstrated that almost 33% of all known peptide hormones may carry O-glycans. This is an over-statement. A firm conclusion can be stated for the four species examined and for the limited number of tissues/fluids examined. The suggestion/hypothesis/speculation can then be made that similar results will be observed in the many other organisms/tissues and peptide families that have not yet been examined. The authors state (p.5) that peptide hormones are classified into 46 distinct families, without making it clear that their statement applies only to the closely related species selected for study, not to all eukaryotic peptide hormones. NeuroPeP, which includes over 400 organisms, lists well more than 46 distinct families.

Response 1: We agree. The sentences on p. 5 has been modified to stress that the data presented here concerns peptide hormones in the four species investigated, and that we speculate that this phenomenon can be extrapolated to other mammalian species. Also, please see reviewer 2 response and Action 4.

Action 1: We modified the following sentence to stress that we are selectively presenting the human peptide hormone families in Fig. 2;

“Human peptide hormones are classified into 46 distinct families of which O-glycosylated members were identified in 27 of the families (Fig. 2 and Supplementary Data 1 & 2). The algorithm NetOGlyc¹⁵ predicted that the majority of peptide hormone families contain glycosylated members (41 out of 46).”

Query 2: To make this body of work accessible to a wider audience, a better introduction to O-glycosylation in general and GalNAc-type O-glycosylation is needed. The symbols used to depict O-glycans (yellow square, yellow circle, etc.) in Fig.3 are not defined until Fig.4b, when the authors finally illustrate the structures about which they have been writing. It is never made clear whether GalNAc-type O-glycosylation accounts for a large or small fraction of O-glycosylation.

Response 2: We agree.

Action 2: The following text has been added to the introduction first paragraph p.4 :

"O-glycosylation is a highly abundant post translational modification estimated to affect 80% of secreted proteins in a cell. The biosynthesis is initiated by a large family of 20 partly redundant isoenzymes (GalNAc transferases) and recently we and others have..."

The O-glycan symbol nomenclature is now introduced in Fig. 3.

Query 3: The authors deal with technical issues about the best ways to identify O-glycosylation sites in peptide hormones and with the results of their analyses. As currently written, these two topics are intermingled - discussion of technical considerations (LMWE strategy vs Rapigest) appears after discussion of conservation across species and location of the sites within propeptides. The manuscript would be much easier to follow if these issues were presented separately. A rationale should be provided for the organisms/tissues/biofluids selected for study. Fig.1 deals with some of the technical issues, but the conclusions are not clearly stated. For example, it is not readily apparent what the 0, 1, 2 and 3 extraction methods referred to are? In Fig.1d, can the authors suggest why 10 glycosylated peptides were missed by their new scheme, but detected using Rapigest or a cell line?

Response 3: The analysis of glycosite conservation across species precedes the description/characterization of the glycopeptide hormone atlas and as such we find it constructive to introduce this analysis first, followed by the technical and qualitative description. We believe we provided a rationale for the selection of tissues in the first paragraph of the results section:

"The strategy (presented in Supplementary Fig. 1a) was applied to both cell lines (N2A and STC-1) and organs known to express and/or secrete high levels of diverse peptide hormones, including porcine and rat brain, cerebellum and pancreas as well as porcine intestinal ileum, and human plasma, cerebrospinal fluid (CSF) and prostate cancer biopsies."

The differential identification across the tissue samples and e.g. cell lines may arise from the differential expression of proteins and peptides in these biological sources, however the major point from Fig. 1d is that the LMWE strategy increases the total identification of glycosylated peptide hormones compared to Rapigest extraction.

Action 3: To further clarify the order of sequence in making the atlas, we modified the 1st paragraph of the results section as detailed in response/action to reviewer 2 query 4.

Furthermore, the extraction methods are now described in more details in the methods section p. 14 and in Suppl. Data 2, and we have added a summary of the used extraction methods to the legend of Figure 1.

"Figure 1. Summary of the identification of glycosylated peptide hormones using LC-MS/MS

a) Number of glycosites identified on mature and propeptide hormone proteins. **b)** Heatmap illustrating the identification of glycans on mature peptide hormones across different biofluids, tissues and cell lines. Greyscale intensity illustrates the number of extraction methods that have resulted in the identification of a given glycopeptide. See supplementary data 3 for examples of extracted spectra of glycopeptides from each peptide hormone. **c)** Bar graph of the total number of mature glycopeptide hormones identified in each tissue or cell line (All extraction procedures combined). **d)** Venn diagram illustrating the distribution of identified glycosylated peptide hormones between the different extraction

methods (total protein extractions (Rapigest extractions and Crude biofluids), low molecular weight extractions (LMWE: H₂O, acetone, acetone/AcOH, AcOH and EtOH/HCl) and cell lines (Supernatant and total cell lysate).”

Query 4: In several instances, the significance of the claims made is not supported by an appropriate statistical analysis. For example, in Fig.4i, error estimates are needed for the half-lives reported, along with statistical tests to establish which differences are significant. Fig.4h reports susceptibility to proteolytic degradation – it is scored as ++ vs + vs -, with no quantitative statement of how + is distinguished from ++. On p.7, the authors state that “increased EC₅₀-values correlated positively with the size of the O-glycan” – the statistical significance of this statement is not clear.

Response 4: We agree.

Action 4: We have carefully revised the statistical analysis supporting the data presented in Fig. 4d, 4e, 4g, 4i, S5a and S5b and modified the Legends as follows:

“Figure 4. Functional analysis of O-glycans within the receptor binding region of peptide hormones.

a) Schematic depiction of the Thr7 O-glycan in the receptor binding domain of the glucagon hormone family (See Supplementary Fig. 7e). **b)** Structure of the sialyl-T glycan. **c)** Schematic depiction of the Thr32 O-glycan in the receptor binding domain of the NPY hormone family³⁸. **d)** Potencies ($\text{Log}_{10}(\text{EC}_{50})$) of glycosylated glucagon, VIP and GLP1 determined in cell based receptor activation assays ($n=3$ independent experiments in duplicate determinations for all groups, except non-glyc. GCG where $n=5$). **e)** Potencies for glycosylated NPY^{Thr32} and PYY^{Thr32} determined and shown as in d) ($n=3$ for all groups). **f)** Western blot using an antibody recognizing mature and proNPY showing high stoichiometry of glycosylated proNPY. Media from NPY transfected STC-1 was either desialylated with neuraminidase (neu), treated with the glycoprotease Operator, or both in combination. **g)** Alpha helical content of 20 μ M NPY or its glycoforms determined by CD-spectroscopy (pH 7 at 25°C). ($n=4$ independent measurements for all groups except Tn where $n=3$). **h)** Summary of in vitro degradation of peptide hormones using purified Neprilysin (NEP) or Dipeptidyl-peptidase IV (DPP-IV) monitored in a MALDI-TOF time-course assay (Supplementary Fig. 8 for spectra). Qualitative scoring ++: full susceptibility, +: reduced susceptibility, -: non-susceptible to proteolytic cleavage compared to non-glycosylated peptide. **i)** Plasma concentrations of GLP-1 or glycoforms after intravenous bolus injection (20 nmol/kg) in anesthetized rats ($n=8$ rats pr group). GLP-1 was measured by an assay targeting the amidated C-terminus of the molecule measuring total GLP-1 (intact 7-36 amide and metabolites of this). Half-lives ($T_{1/2}$) are indicated with 95% confidence intervals. For each glycoform, only timepoints represented by $n>4$ were included in the analysis ($n_{(\text{GLP-1})}=35$, $n_{(\text{GLP-1 T})}=38$, $n_{(\text{GLP-1 ST})}=41$ measurements). Statistics: All error bars represent mean \pm S.E.M.. Two-tailed one-way ANOVA (Tukey’s post-hoc test) was performed in panels e), d), f), except for comparisons within NPY2R+PYY-, NPY5R+PYY- and NPY5R+NPY-assays in panel d) where students t-test was performed. Multiple linear regression was performed in panel i). ns, not significant; * $p<0.05$; ** $p<0.01$; *** $p<0.001$. Source data for panel d), e), f), h), i) are provided as a Source Data file.”

and

“Figure S5. Receptor activation of peptide hormones and their glycoforms at concentration fixed at 1 μ M.

a) Receptor activation of VIP, Glucagon and GLP1 and their Thr7 glycosylated forms determined in cell based receptor activation assays. 100% receptor activation is defined as the maximal response (E_{max}) obtainable with the cognate non-glycosylated agonist. Data points are shown as as mean +/- S.E.M from three independent experiments performed in duplicate assays. (Apart from non-glycosylated GCG that was done in 5 independent experiments). **b)** Receptor activation of 1 μ M NPY and PYY in their Thr32 glycoforms performed as described for a). (* $p < 0.05$; ** $p < 0.01$, *** $p < 0.001$, two-tailed one way ANOVA with Tukey’s post-hoc test). Source data for panel a) and b) are provided as a Source Data file.”

Query 5: 1. Methods: lectin beads were typically eluted with the appropriate sugar and then with that same sugar at double the concentration – it is not clear whether the eluates were pooled or analyzed separately?

Response 5: We agree.

Action 5: This has been clarified by adding the following sentence in the Methods section:

“5% of the elution fractions were analysed by LC-MS to check for sample complexity. If appropriate, the elution fractions were pooled, and fractionated using either high pH-fractionation or IEF prior to nLCMS/MS analysis.

Query 6: p.17: STC-1 cells were stably transfected with preproNPY. A vector encoding preproNPY was used to transfect the cells – what vector was used? what species of proNPY did it encode? what was the source of the vector?

Response 6: Agree, this information was left out.

Action 6: Following sentence has been added to the Methods section p.18:

“PreproNPY-plasmid (Human, Uniprot: P01303) was obtained from genewiz in the pUC57 framework and subcloned into pcDNA3-plasmid (Invitrogen, Thermo Fischer Scientific) with Bam/NotI restriction enzymes (New England Biolabs) followed by ligation with T4 DNA ligase (Thermo Fisher Scientific).”

Query 7: The sources of many materials are not provided.

Response 7: Agree.

Action 7: We have carefully checked the Material and Methods section and added appropriate information.

Query 8: When the NeuroPeP website is mentioned, reference 24 is cited; the actual website should be provided.

Action 8: An address for the website is now provided after first mention of NeuroPeP:

“We designed a proteomics-based workflow targeting low-molecular weight glycoproteins with different extraction protocols and lectin enrichments (LMWE)^{15,16,23}, and applied a comprehensive database of neuropeptides and peptide hormones (NeuroPep; <http://isyslab.info/NeuroPep/>)²⁴ for informatics.”

Query 9: The reader is never told anything about the two cell lines included in the study (N2A and STC-1) – What is their derivation? How were they obtained? Why were they used?

Response 9: We agree that the information provided is limited. Regarding motivation for choosing these cells we have already stated (results-section 1. paragraph p.4) that STC1 and N2a were chosen due to their high expression of peptide hormones.

Action 9: We have included the following sentence: Methods p.13:

“Cell protein extraction. STC-1 cells derived from mouse invasive small intestinal neuroendocrine carcinoma and N2a cells from mouse invasive small intestinal neuroendocrine carcinoma were obtained from ATCC and maintained in DMEM (Sigma) media supplemented with 10 % fetal bovine serum (Sigma) and 4 mM GlutaMAX (Gibco) at 37°C and 5 % CO₂....”

Query 10: In Fig.4f, the authors use a STC-1 cell line expressing proNPY to explore the extent to which proNPY is O-glycosylated. Two bands are clearly visible on the Western blot and collapse into one after neuraminidase treatment; no explanation is given for the location of this band at a position intermediate to the two bands in the untreated sample. Treatment with neuraminidase followed by OperATOR O-glycoprotease produces a slight downward shift in mobility, from which the authors conclude that there are two distinct glycoforms. These experiments are presented without sufficient interpretation/explanation. Not every reader will know what OperATOR does. It is impossible to interpret the size shift following OperATOR cleavage without guidance from the authors about the expected size shift and the specificity of the NPY antibody being used – would its ability to detect proNPY be affected by OperATOR cleavage? Is the reader to conclude that all of the proNPY is O-glycosylated?

Response 10: We agree that the Operator treatment might be challenging to interpret.

Action 10: Please see Reviewer 1 Query 4 Response and Action.

Query 11: On p.9, the authors state that “Both DPP-IV and NEP degraded all of the tested non-glycosylated peptides fully or partially within 15-30 min ...”. Based on the Methods section (p.19), the enzyme concentrations were adjusted to achieve this goal, with very different amounts of each protease used for different propeptides. Although it is clear that the glycopeptides are more stable, the conclusion that the protective effects of the O-glycans increases with the size of the O-glycan structure requires statistical analysis.

Response 11: We agree, the qualitative scoring system applied to the degradation assays does not allow for statistical analysis.

Action 11: We have modified the statement (2nd paragraph p.9) which now reads “ In some cases, the intact glycopeptide hormones were still detectable even after overnight incubations. The protective effects seemed to be most profound with the larger glycan T- and sialyl-T isoforms”

Minor points: there are many typographical errors throughout the text

Query 12: p.5 – third line from bottom – the text ending with reference 27 is not a sentence.

Action12: Fixed, thank you.

Query 13: p.9 – 2nd line before the Discussion – “As a results”, instead of “As a result”

Action 13: Fixed, thank you.

Query 14: P.13: 0.5 M CH₃COOH were added icecold – presumably “were added to ice cold”?

Action 14: Fixed, see reviewer #3, action 3

Query 15: Throughout – abbreviations should be defined at first use and their use minimized (e.g., Rapigest, LWAC, MQ)

Action 15: Fixed, thank you.

Query 16: p.14: line 2, “as done for the the LMWF-enriched”

Action 16: Fixed, thank you.

Query 17: p.14: line 3, “icecold PBS, scrabing off” – presumably “ice cold PBS, scraping off”?

Action 17: Fixed, thank you.

Query 18: p.17 – lines 2&3 of “Purified glycopeptides.” section – “the VIP and secretin peptides which was only” – verb should be plural, “were”. Later in same paragraph, “The Tn-glycosylated peptides was further” – “were”

Action 18: Fixed, thank you.

Query 19: p.18: “6 mio cells/flask”??

Response 19: We agree that this statement is unclear. This number refers to the seeding density of the COS-7 cells

Action 19: We changed the sentence as follows “seeding density of 6 million cells per T175 flask”

Query 20: If the “structures” shown in Fig.4a and Fig.4c are not based on actual data for these ligand/receptor interactions, it is not clear that adding a cartoon of the sugar chain increases understanding.

Response 20: Agree. As stated in the paper, crystal structures of both Glucagon-receptors and NPY-R1 have been solved which illustrates that the position of the glycans indeed are in flanking regions of the receptor-interacting part of the hormones.

Action 20: Supplementary Figure 5 has been expanded with panel e) showing the crystal structure of glucagon (with highlighted glycosylated residue) interacting with its receptor and a citation to the NPY1R crystal structure and modelling of NPY-interaction is added to the legend of Figure 4, to support the illustrations in Figure 4a and 4c (see reviewer 3 action 4).

Query 21: The NetOGlyc algorithm is mentioned, with citation of reference 15; the website used should be indicated.

Action 21: Fixed, thank you.

Query 22: Fig. S1 – PNA, JAC, VVA lectins – specificity should be stated.

Action 22: Fixed, See Reviewer 1, Action 7.

Reviewers' comments:

Reviewer #1 (Remarks to the Author):

The authors have mostly addressed my concerns and question. However, I still have some concerns regarding the presentation of the experiments and results and about the usefulness of the experiment aiming the demonstration of the high extent of glycosylation of the NPY propeptide. These should be addressed before acceptance for publication.

1. Original query 1 of reviewer 1

Unfortunately, the presentation of the identifications is still unacceptable. The data provided is still inappropriate for the reader to identify the spectra listed in Sppl. Data 2. There are no ACTUAL raw file names only the mgf and msf file names but these seem to be the originating from multiple raw files. Spectrum identifiers, m/z values, retention times and XCorr scores refer to only the „first” identification whatever that should mean. The authors uploaded 1147 files to the Pride repository but these files cannot be sorted either by name or file extension therefore all files must be scanned to find the data desired. In summary, checking the quality of the identifications did not get any easier compared to the original submission. The authors should address this issue before acceptance of the manuscript.

As an illustration, all non-TMT insulin B chain identifications are reported from „151112_Fusion_(LC2)_Lasse_Pancreas_(Nov2015)_2h_HCD_ETD_10%_PNA_15_19(p1)_(1G)_1xT.mgf”. there are 2 PSMs for GFFYPTK+HexNAcHex and 4 PSMs for GFFYPTKA+HexNAcHex. The mgf file seems to refer to 3 raw files, 151112_Fusion_(LC2)_Lasse_Pancreas_(Nov2015)_2h_HCD_ETD_10%_PNA_15_16.raw, 151112_Fusion_(LC2)_Lasse_Pancreas_(Nov2015)_2h_HCD_ETD_10%_PNA_17_18.raw, and 151112_Fusion_(LC2)_Lasse_Pancreas_(Nov2015)_2h_HCD_ETD_10%_PNA_19.raw.

The specified „first scan” numbers do not refer to any MS2 spectra acquired on the precursors m/z specified in Sppl. Data 2. There are 2 PSMs each with the very same First Scan, RT [min], Xcorr, Charge, m/z [Da] and MH+ [Da] values. How should the reader interpret these data?

2. Original query 4 of reviewer 1

Regarding the demonstration of the high extent of O-glycosylation using the NPY prepropeptide – I am still not convinced about the usefulness of these experiments and I think that this part should be omitted from the paper. What are the conclusions here? What evidence supports them? If we presume that all NPY glycoforms separate on the gel (I am not sure it is true as one NeuAc unit induces a <2% change in the molecular mass of the propeptide), then we should conclude that 1. all

detected NPY is glycosylated – otherwise there should be at least 1 additional signal both in the nontreated and neuraminidase-treated samples that is unaltered by neuraminidase treatment 2. Operator is not very efficient for cleaving the peptides though neuraminidase treatment facilitated Operator cleavage (see suppl fig 6, operator alone or combined with neuraminidase treatment). (In fact, it is not very surprising as Operator does not cleave efficiently elongated O-glycans.) 3. Site-specific glycosylation information cannot be deduced from these experiments (except that the antibody recognizes the N-terminal part of the propeptide). Everything else is speculation and the authors do not draw any conclusions from the results except that there must be mucin-type glycosylation in the samples otherwise Operator would not work. So what is the take home message here?

Furthermore I think the authors might misinterpret their results. On Figure 4f the authors show that upon neuraminidase treatment the 2 NPY-related bands collapse into one band and these positions are labelled as „glyc” nad „non-glyc” indicating that the lower band corresponds to a nonglycosylated version. However, neuraminidase removes only the terminating sialic acid units, not the whole glycan! Or do the authors argue that one of the glycans is mono/multisialic acid only? (The same holds for Suppl. Figure 6. as well.)

Minor issues in this part

p. 8.: „Both glycosylation of Thr32 and Ser40 or 41 would result in a fragment of approximately 4-5 KDa after OperATOR digestion”. Positions (32, 40, 41) refer to the processed sequence (mature hormone), while in Suppl. Fig 6 the positions are specified as unprocessed (60, 68, 69).

on p. 8., the authors claim that „The LWAC-based glycopeptide enrichment strategy employed here does not allow for evaluation of the stoichiometry of O-glycosylation at the sites identified. Determination of stoichiometry is a challenging task because of a relatively low ionization of glycopeptides compared to the corresponding peptides³⁹, which is further complicated by the low abundance of peptide hormones in general.” While all true, it sounds as if different ionization efficiency would be the result of the LWAC sample prep. LWAC should enrich glycopeptides so estimation from LWAC samples is impossible (nonglycosylated peptides are diminished/missing). But even if the nonglycosylated peptides would be present, estimation of the stoichiometry would be imprecise due to the different efficiency of ionization and different elution times. Please reword.

3. It is very difficult to follow the different extraction, enrichment and digestion steps (as reviewer 2 also pointed out). Please include a table that defines all combinations you tested – extraction method, lectin type and enzymes used for digestion. Further justification of the experimental conditions and discussion of the results as related to the different experimental methods would also be welcome – this would be very helpful for the readers to design their own experiments (for example, which enzyme/enzyme combinations yielded the best results etc).

Minor issues:

Why do the authors refer to glycopeptides as „glycopeptide fragments” throughout the text? As sample preparation included proteolytic digestion, all identified peptides should represent a larger polypeptide/protein. It would be best to keep „fragment” for fragment ions in an MS-based manuscript.

p. 4. „The biosynthesis is initiated by a large family of 20 partly redundant isoenzymes

80 (GalNAc transferases (GalNAc-T’s))”... Please correct. There are different types of O-glycosylation, the above sentence is true only for mucin-type glycosylation.

p.5. „Of these, 34 sites were identified on 16 human prohormone sequences (Supplementary Fig. 1b) and 386 glycosites were identified on 102 non-human (porcine or rodent) propeptide hormones.” – the numbers in Suppl. Fig. 1 are 384+36.

„Surveying the distribution of the identified human and conserved O-glycosites across each peptide hormone precursor revealed that about half (207) of the glycosites were located in the proprotein part and half (177) of these were located on the mature peptide hormones (Fig. 1a and Table 1).”

Please revise. This sentence suggests that the 177 sites are the subpopulation of the 207 sites mentioned first.

„177 glycosites were distributed on 91 mature peptide hormone sequences” – Fig 1d suggests 93 glycosylated peptide hormones. Please clarify.

„We analysed glycosite locations relative to protein topology and known functional features, and found that while the majority of sites were distributed among regions of the peptide hormones typically referred to as address and message regions” – delete „while”

p. 9. „In some cases the intact glycopeptide hormones were still detectable even after overnight incubations.” – (First, this sentence is included twice.) Please explain. What is „intact” here? I guess the authors refer to the peptide decorated with the largest glycan. However, these are not necessarily the native hormones. I have some further problems with the stability studies and I should have noticed them in the first round. Nonetheless, I would like to know why the specified glycoforms were used for the stability assays. Secretin was detected glycosylated on T7, S8 and/or S11, the peptide used for the study was singly glycosylated. More importantly, the GLP-1 glycoform tested is doubly glycosylated (T5 and T7), while the authors identified only the singly glycosylated version. How relevant are these results for the native hormones? Minor issue: the experimental part seems to be incomplete. While stability assay results are presented for PYY, the „purified glycopeptides” part on p. 19. does not indicate chemical or chemoenzymatic synthesis of the ST

glycoform. Connecting to these studies, in the discussion the authors say (P. 12.): „The most prominent effects were observed with the naturally occurring sialylated ST O-glycan structure, where e.g. PYYThr32 and GALSer23 remained intact even after 24 hrs incubation with neprilysin.” Suppl. Fig 8b clearly shows that both GAL and PYY undergo some degradation and does NOT stay intact after 24-h neprilysin treatment. Please correct.

p. 10. The Discussion part starts with the following sentence: „Here, we developed a strategy to explore O-glycosylation on peptide hormones, and demonstrate that almost 33 % of all known peptide hormones may carry O-glycans.” I think this sentence need rewording as pointed out by Referee 3 in query 1 and stick to the species/organs investigated.

p. 10. line 283: correct „homones” to hormones

p. 19. line 563: replace „over night” with overnight

Reviewer #2 (Remarks to the Author):

The authors have addressed the majority of concerns raised in the initial review. A few additional concerns remain:

1. Page 8, lines 222-223 do not make sense and need to be revised: “When treating with OPERATOR the in addition to the upper band lost intensity, a band of approximately 4-5 KDa appeared (Supplementaty Fig. 6a).” Note that Supplementary is mis-spelled.

2. The legend for new Supplementary Figure 6b is also confusing and needs to be revised: “The theoretical mass-change by cleavage at T-glycosylated Ser3 will show approximately same mass as full length non-glycosylated proNPY (+0.1kDa) due to the size of the glycan on the remaining antibody-recognized fragment.”

3. The revised legend for Figure 1b refers to “supplementary data 3”, but the Supplementary Data Excel Spreadsheet only has tabs for Supplementary Data 1, 2, and 4.

4. The title for new Supplementary Figure 2 does not make sense and needs to be revised: “illustrates selected peptide hormone precursors their identified O-glycosylation sites relative to the proteolytic activation sites of peptide hormones.”

Reviewer #3 (Remarks to the Author):

Faced with lengthy, but very positive, critiques from all three reviewers, the authors made multiple modifications to their manuscript, making it much more accessible to the diverse audience that should benefit from the information provided. The extensive O-glycosylation of peptide hormones documented by the authors will have a profound effect on subsequent studies, on the development of peptide-based pharmaceuticals and on clinical assays for peptide hormones. The scope of the study is large, and the inclusion of structural studies, half-life studies and receptor activation studies reinforces the importance of the observations made.

That said, there are many little details that still need to be fixed. I have noted quite a few, but surely did not identify all of the singular/plural errors and repeated words. This work is too good to have it compromised by easily fixed errors.

Suggestions:

1. Lines 67-68 – a few peptide hormones (e.g. calcitonin and ANP) are not “stored in secretory vesicles awaiting” – just say “most peptide hormones are stored”
2. In modifying their discussion of stoichiometry, the authors made this section very long. Since OPERATOR was of no use on the TCA-precipitated sample, why not show only neuraminidase and move the OPERATOR data for HEK293 produced proNPY to Fig.4? The “reduced intensity” argument is less than compelling and should just be deleted. Much of what appears in Results could move into the Figure legend, keeping the focus on stoichiometry.
3. In Fig.4h there are 8 test peptides/proteases – in 7 of 8 cases, the data for Tn and T are identical. The data for ST differ from the data for T in 7 of 8 cases. The final sentence of this section should not

group the protective effects of the T- and sialyl-T isoforms together; it is only the sialyl-T isoform that has a more profound effect. This comment is relevant to line 309 in the Discussion, where the authors state that “the size of the O-glycan correlated positively with the observed reduction in potency (Fig.4)” – this statement should be modified to more accurately reflect the data presented.

4. In discussing the half-life data in Fig.4i, the authors group GLP-1Thr7-T and GLP-1Thr7-ST together (total GLP-1Thr7) and then present two clearly different half-lives (6.4 min and 22.7 min, respectively) – for this sentence to be understandable, the two glycoforms need to be specified, not “total glycoforms”.

5. Lines 333-334 – the two peptidomics studies (“both”) referred to here are not clear – please clarify.

6. The widespread occurrence of O-glycosylation may also alter the ability of widely used clinical assays to detect these various peptide hormones, leading to discrepancies between laboratory data and clinical observations; this aspect of extensive O-glycosylation might be worth mentioning.

Minor edits

Line 35 – “peptide hormones”, not “peptides hormones”

Line 72 – “concentration ... has challenged”, not “have challenged”

Lines 139-141 – This is not a sentence; delete “while” on line 140?

Fig.2 line 969 – Lettering is cut off. This legend is not well written.

Line 970 – “illustrate that 27” – “illustrated that 27”?

Line 974 – “sites are also predicted glycosylated” – “sites are also predicted glycosylation sites”?

Line 975 – the graph does not show the “proportion of identified sites”, it shows the number of sites.

Line 975 – “NetOGlyc 4.0 do not predict” – “NetOGlyc 4.0 does not predict”.

Line 185 – “EC50-value of 200 nM EC50 compared” – should likely be “EC50-value of 200 nM compared”

Line 192-193 – sialic acids – plural; should be singular to go with “its strong negative charge”

Line 217 – “induced by neuraminidase” should be “produced by neuraminidase”

Lines 263-264 – The sentence starting with “In some cases ...” is repeated

Line 326 – need to make it clear that superscript P (XP) and superscript G (XG) indicate phosphorylation and glycosylation, respectively.

Line 345 – “capping of glycans are needed” – should be “is needed”

Lines 426-427 – medium/media – the term is repeated when it should not be

Line 530 – Genewiz should be capitalized and a website or location given

Line 538 – polysciences – presumably a company with a location?

Line 563 – Cell signaling is Cell Signaling Technologies

Reviewers' comments:

Reviewer #1 (Remarks to the Author):

The authors have mostly addressed my concerns and question. However, I still have some concerns regarding the presentation of the experiments and results and about the usefulness of the experiment aiming the demonstration of the high extent of glycosylation of the NPY propeptide. These should be addressed before acceptance for publication.

Query 1 - Original query 1 of Reviewer 1:

- a) Unfortunately, the presentation of the identifications is still unacceptable. The data provided is still inappropriate for the reader to identify the spectra listed in Sppl. Data 2. There are no ACTUAL raw file names only the mgf and msf file names but these seem to be the originating from multiple raw files. Spectrum identifiers, m/z values, retention times and XCorr scores refer to only the „first” identification whatever that should mean. The authors uploaded 1147 files to the Pride repository but these files cannot be sorted either by name or file extension therefore all files must be scanned to find the data desired. In summary, checking the quality of the identifications did not get any easier compared to the original submission. The authors should address this issue before acceptance of the manuscript.
- b) As an illustration, all non-TMT insulin B chain identifications are reported from „151112_Fusion_(LC2)_Lasse_Pancreas_(Nov2015)_2h_HCD_ETD_10%_PNA_15_19(p1)_(1G)_1xT.mgf”. there are 2 PSMs for GFFYPK+HexNAcHex and 4 PSMs for GFFYPKA+HexNAcHex. The mgf file seems to refer to 3 raw files,
151112_Fusion_(LC2)_Lasse_Pancreas_(Nov2015)_2h_HCD_ETD_10%_PNA_15_16.raw,
151112_Fusion_(LC2)_Lasse_Pancreas_(Nov2015)_2h_HCD_ETD_10%_PNA_17_18.raw, and
151112_Fusion_(LC2)_Lasse_Pancreas_(Nov2015)_2h_HCD_ETD_10%_PNA_19.raw.
The specified „first scan” numbers do not refer to any MS2 spectra acquired on the precursors m/z specified in Sppl. Data 2. There are 2 PSMs each with the very same First Scan, RT [min], Xcorr, Charge, m/z [Da] and MH+ [Da] values. How should the reader interpret these data?

Response 1:

- a) We realize that the MSMS spectra evaluation was not sufficiently transparent. We have modified Suppl Data 3 and additionally supported each entry with the corresponding raw file names. Also, we have substituted the PRIDE server for the Massive Server (Massive.ucsd.edu) allowing us to group files.
- b) We regret that the reviewer could not find the corresponding raw MSMS spectrum of non-TMT insulin B chain identifications. The reason likely was that the m/z values from the .mgf entries are glycan mass subtracted. We have explained this subtraction strategy as well as the processing mechanism of using glycan-subtraction in details previously (Vakhrushev et al., Molecular and Cellular Proteomics 2013) and also briefly in the current manuscript. Due to high degree of glycan neutral losses the most intensive fragments in the MSMS spectrum do not have the glyco PTM anymore. Currently the Sequest search engine cannot address this issue properly. To improve identification score (typically by factor of two) we previously introduced a method to apply combinatorial subtraction of a defined glycan mass from the precursor ions and add subtracted epitope as a tag to the subtracted file for later identification. Thus, glycopeptides are ”converted to non-glycosylated”. This procedure increases the identification depth. Thus, in order to find the corresponding precursor ions, the sugar mass increment has to be added back to the m/z listed in the table.

Action 1:

- a) We have substituted the PRIDE server for the MassIVE Server (Massive.ucsd.edu) under dataset identifier MSV000085289, which allows for files to be grouped in folders by filetype, biosource and extraction method. This allows for easy navigation through the files, where single files can be downloaded directly in the browser at the MassIVE website. Entire folders with complete sets of .raw, .mgf, or .msf files for any sample can be downloaded by FTP-link. To ensure that data is easily accessible for the reviewers, we have included a guide on how to access the files both through the browser as well as through connection to the FTP-server (we have ensured that the download works on multiple computers including both mac and windows machines). When data is made publicly available, it will be accessible without the requirement of login, and thus the provided guide will be irrelevant for the individual reader. Additionally, upon making the dataset public, all data will be uploaded to the ProteomeXchange server under the dataset identifier: PXD018560

Furthermore, we have reprocessed the data so that each .mgf file now only refers to one .raw file instead of multiple. Moreover, we have expanded Supplementary Data 3 to include columns that tabulates individual identification entries with the corresponding .msf, .raw and potential .mgf files. This approach should now make it easier for the reader to find any and every annotated MSMS spectra through Proteome Discoverer .msf files by searching the corresponding sequence and scan number. In addition, we have removed the search for NeuAc in the case of CSF and Heart in order to improve consistency (note that peptide hormone Neuromedin-B was only identified with NeuAc-decorated glycans and is thus removed from Figure 3 and Figure S3). The reprocessing has led to slight changes in the total number of identified glycosites (which has been updated in Supplementary Figure 1b), however, this has no impact on any conclusions drawn from the study.

- b) To make sure that the glycan mass subtraction of HCD spectra is totally clear for the reader, we have added columns including both m/z and MH+ with the glycan mass added back. For absolute clarity we have retained the glycan-subtracted m/z and MH+ used for identification in columns labeled “Glyco subtracted m/z” and “Glyco subtracted MH+”. The subtraction is only performed for HCD-fragmented PSMs, and is thus not determined for ETD-PSMs (marked as ND). An additional “Activation” column has also been added to clarify which PSMs were sequenced with either ETD or HCD. The Supplementary Data has been thoroughly checked for duplicate PSMs, which has now been removed.

Query 2 - Original query 4 of Reviewer 1: Regarding the demonstration of the high extent of O-glycosylation using the NPY prepropeptide – I am still not convinced about the usefulness of these experiments and I think that this part should be omitted from the paper. What are the conclusions here? What evidence supports them?

Response 2: This comment must rely on a misunderstanding – stoichiometry is an important factor when considering O-glycosylation. However, this is also one of the greatest challenges for the field to day. Providing this data for one example of the many identified glycopeptides we believe is highly valuable, and this was/is extensively discussed in the Discussion. The importance of stoichiometry was also raised by Reviewers 2 and 3 in the first review phase.

Action 2: We have rearranged Supplemental Figures 6a and 4 to include both the STC-1 and HEK293 immunoblots, and removed the OperATOR data for the STC-1 produced proNPY as suggested by Reviewer #3 (query 2). The paragraph on pro-NPY glycosylation p.8-9 lines 273-347 of the results, has been modified to clearly stress these conclusions as follows:

“Direct SDS-PAGE Western blotting of trichloroacetic acid concentrated conditioned media showed that secreted proNPY migrated as two distinct and equally intense bands (Fig. 4f, lane 1). Treatment with

neuraminidase to remove sialic acids on glycans collapsed the two immunoreactive bands of proNPY into one apparently co-migrating band, which demonstrated that the upper band of proNPY contains sialylated glycans (lane 2). To confirm the presence of GalNAc-type O-glycans, we expressed proNPY in a high producer cell line, HEK293 6E, that secreted proNPY in amounts directly detectable by western blotting without concentrating the sample further. HEK293 6E produced proNPY migrated as three bands; two upper strong and one lower weak band (Fig. 4g, lane 1). As with STC-1, neuraminidase demonstrated that a considerable amount of proNPY contains sialylated glycans (Fig. 4g, lane 2). For further confirmation we used a recently characterized O-glycoprotease from *A. muciniphila*, OpeRATOR, which specifically cleaves O-glycopeptides immediately N-terminal to an O-glycan, but is blocked by the presence of sialic acid⁴⁰. Treatment with OpeRATOR only in combination with neuraminidase resulted in the loss of the intensity of the upper band concomitantly with the appearance of a band migrating approximately as 4-5 kDa (Fig. 4g, lane 3) confirming that a majority of proNPY contains one or more O-glycans with sialic acid. In our atlas, we identified multiple glycosylation sites on proNPY, and based on this data it was not possible to determine the specific OpeRATOR cleavage site(s) (Supplementary Fig. 6). However, the data confirmed that two mammalian cell lines, STC1 and HEK293 6E, produce proNPY with at least 50 % sialylated GalNAc-type O-glycans.”

Query 3: If we presume that all NPY glycoforms separate on the gel (I am not sure it is true as one NeuAc unit induces a <2% change in the molecular mass of the propeptide), then we should conclude that 1. all detected NPY is glycosylated – otherwise there should be at least 1 additional signal both in the nontreated and neuraminidase-treated samples that is unaltered by neuraminidase treatment.

Response 3: This does not appear to be an insightful comment? The effects of sialic acids (NeuAc) on SDS-PAGE mobilities are well known to be unpredictable and unrelated to the actual mass, and here we are talking about small peptides and the charge effects may be much greater than seen with larger glycoproteins. We can certainly agree that all glycoforms may not be resolved, but the interpretation that a major part of pro-NPY is sialylated (and O-glycosylated) is the take-home message and what is concluded.

Action 3: We have clarified this further in the text. Please see Reviewer 1, action 2

Query 4: Operator is not very efficient for cleaving the peptides though neuraminidase treatment facilitated Operator cleavage (see suppl fig 6, operator alone or combined with neuraminidase treatment). (In fact, it is not very surprising as Operator does not cleave efficiently elongated O-glycans.)

Response 4: See response 2.

Action 4: See action 2.

Query 5: Site-specific glycosylation information cannot be deduced from these experiments (except that the antibody recognizes the N-terminal part of the propeptide). Everything else is speculation and the authors do not draw any conclusions from the results except that there must be mucin-type glycosylation in the samples otherwise Operator would not work. So what is the take home message here?

Response 5: See response 2.

Action 5: See action 2.

Query 6: Furthermore I think the authors might misinterpret their results. On Figure 4f the authors show that upon neuraminidase treatment the 2 NPY-related bands collapse into one band and these positions are labelled as „glyc” nad „non-glyc” indicating that the lower band corresponds to a nonglycosylated version. However, neuraminidase removes only the terminating sialic acid units, not the whole glycan! Or

do the authors argue that one of the glycans is mono/multisialic acid only? (The same holds for Suppl. Figure 6. as well.)

Response 6: This is again does not appear to be an insightful comment as related to the Reviewers query 3. It is well known in the field that sialic acids have a major effect (although unpredictable) on SDS-PAGE migration, but whether the neutral part of a small O-glycan here clearly predicted to be only core1 (Gal β 1-3GalNAc) affects the mobility enough to be resolved is unclear. What is clear is that neuraminidase treatment collapses two migrating bands to one, and we simply conclude in accordance with customs in the field that the band that changes migration is glycosylated. Since this is most likely a core1 O-glycan (this is inferred from the lectin isolation) this may have variable 1-2 sialic acids.

Action 6: See response 2.

Minor issues in this part

Query 7: p. 8.: „Both glycosylation of Thr32 and Ser40 or 41 would result in a fragment of approximately 4-5 kDa after OPERATOR digestion”. Positions (32, 40, 41) refer to the processed sequence (mature hormone), while in Suppl. Fig 6 the positions are specified as unprocessed (60, 68, 69).

Response 7: Thank you for help with clarifying this.

Action 7: Suppl. Fig 6 now includes mature peptide hormone numbering in parenthesis.

Query 8: on p. 8., the authors claim that „The LWAC-based glycopeptide enrichment strategy employed here does not allow for evaluation of the stoichiometry of O-glycosylation at the sites identified. Determination of stoichiometry is a challenging task because of a relatively low ionization of glycopeptides compared to the corresponding peptides³⁹, which is further complicated by the low abundance of peptide hormones in general.” While all true, it sounds as if different ionization efficiency would be the result of the LWAC sample prep. LWAC should enrich glycopeptides so estimation from LWAC samples is impossible (nonglycosylated peptides are diminished/missing). But even if the nonglycosylated peptides would be present, estimation of the stoichiometry would be imprecise due to the different efficiency of ionization and different elution times. Please reword.

Response 8: We thank the Reviewer for the suggestion

Action 8: We have further clarified this p. 8 as follows:

“Estimation of stoichiometry using LC-MS/MS is difficult with LWAC enriched glycopeptide samples, because the non-glycosylated peptide fraction is depleted by the lectin enrichment. In addition to this, the determination of glycosylation stoichiometry in non-enriched samples is challenged by the general sample complexity and dynamic range³⁹. Studying the low abundant peptide hormones that are often not detectable in proteomics studies further adds to the difficulties in detection of glycopeptides let alone identification of glycoforms and stoichiometry of glycosylation. Thus, to address stoichiometry with one peptide hormone...”

Query 9. It is very difficult to follow the different extraction, enrichment and digestion steps (as reviewer 2 also pointed out). Please include a table that defines all combinations you tested – extraction method, lectin type and enzymes used for digestion. Further justification of the experimental conditions and discussion of the results as related to the different experimental methods would also be welcome – this would be very helpful for the readers to design their own experiments (for example, which enzyme/enzyme

combinations yielded the best results etc).

Response 9: We agree.

Action 9: We have now added Supplementary Data 2 listing extraction methods as well as enzymes and lectin enrichment strategies. A discussion of the results has been added to the Results (p. 5) as follows:

“In general, the LMWE resulted in additional identifications of peptide hormone O-glycosites compared to the traditional analysis of total tissue/cell extracts (Rapigest) and biofluids, and in total contributed with more than 1/4 (24) of the peptide hormones identified with O-glycosylation (Fig. 1d). Each LMWE resulted in the identification of a similar number of glycosylated peptide hormones per biosource analysed (Supplementary Data 2) where subtle qualitative differences were noted. For example, glycosylated IGF-II was only identified in CSF after EtOH extraction, glycosylated GLP-1 was found in ileum only after acidic extraction, and glycosylated CCB peptide was observed only after acetone extraction in both whole brain and cerebellum.”

Minor issues:

Query 10: Why do the authors refer to glycopeptides as „glycopeptide fragments” throughout the text? As sample preparation included proteolytic digestion, all identified peptides should represent a larger polypeptide/protein. It would be best to keep „fragment” for fragment ions in an MS-based manuscript.

Response 10: The use of “glycopeptide fragments” was implemented as an attempt to distinguish these glycopeptide (fragments) from the native (glyco)peptide hormones. However, we can agree with the reviewer comment.

Action 10: We have modified the text accordingly and now refer to MS identified glycopeptides merely as “glycopeptides”.

Query 11: p. 4. „The biosynthesis is initiated by a large family of 20 partly redundant isoenzymes

80 (GalNAc transferases (GalNAc-T’s))”... Please correct. There are different types of O-glycosylation, the above sentence is true only for mucin-type glycosylation.

Response 11: Thank you, we inadvertently left out this specification.

Action 11: We have reworded the sentence (p. 4) as follows:

“Mucin-type O-glycosylation (hereafter simply O-glycosylation) is a highly abundant PTM estimated to affect 80 % of secreted proteins in a cell.”

Query 12: p.5. „Of these, 34 sites were identified on 16 human prohormone sequences (Supplementary Fig. 1b) and 386 glycosites were identified on 102 non-human (porcine or rodent) propeptide hormones.” – the numbers in Suppl. Fig. 1 are 384+36.

Response 12: Thank you

Action 12: We have revised Supplementary Figure 1b and legend to facilitate the reading of the main text. Note that due to the reprocessing mentioned in query 1, action 1, numbers have changed slightly

Revised Figure S1b legend: “Supplementary Figure 1. Workflow for identification of glycosylated peptide hormones. a) Proteins from biofluids (cerebrospinal fluid, plasma), neuroendocrine tissue (brain, cerebellum, pancreas, ileum, heart and prostate) and neuroendocrine cell lines (STC-1, N2a) were extracted using up to three different extraction procedures per sample (see Supplementary Data 3). Subsequently, proteins were reduced, alkylated and digested with either trypsin, GluC or chymotrypsin followed by de-sialylation using neuraminidase and glycopeptide enrichment by LWAC using either PNA, JAC or VVA lectins. Glycopeptides were fractionated using either isoelectric focusing or high pH-fractionation before separation and sequencing by LC-MS/MS. The resulting O-glycoproteome was matched against the NeuroPep database and resulted in identification of 445 glycosites on peptide hormones from all species analysed. **b)** Schematic representation of the data analysis of glycosylated peptide hormones. Using sequence alignment, 374 glycosites in peptide hormone orthologs (pig, rat and mouse) were predicted to be conserved in humans based on the conservative preservation of Ser/Thr residues within +/- 5 amino acids. In total 36 human and 374 conserved glycosites were included for further analysis. Mapping glycosites to full peptide hormone precursor sequences revealed that 223 glycosites were located in pro-domains and 187 glycosites were located on mature peptide hormones. In total, 91 mature peptide hormones contained at least one glycosite (See supplemental data 1). *) A large fraction of glycosites located in pro-domains (181 out of 223) are found on the granin, nucleobindin and kininogen superfamilies. **d)** Schematic depiction of the biosynthetic pathway of the most common O-GalNAc-type glycans present on proteins. The majority of circulating O-glycoproteins carry the Sialyl-T (ST) and di-sialyl-T (diST) structures”

Query 13: „Surveying the distribution of the identified human and conserved O-glycosites across each peptide hormone precursor revealed that about half (207) of the glycosites were located in the proprotein part and half (177) of these were located on the mature peptide hormones (Fig. 1a and Table 1).”
Please revise. This sentence suggests that the 177 sites are the subpopulation of the 207 sites mentioned

first.

Response 13: Agree

Action 13: Corrected, the sentence now reads as follows:

“Surveying the distribution of the identified human and conserved O-glycosites across each peptide hormone precursor revealed that about half (223) of the 410 glycosites were located in the proprotein part, and the other half (187) were located on the mature peptide hormones (Fig. 1a and Table 1).”

Query 14: „177 glycosites were distributed on 91 mature peptide hormone sequences” – Fig 1d suggests 93 glycosylated peptide hormones. Please clarify.

Response 14: Agree.

Action 14: Data has been reanalysed and Figure 1d has been changed accordingly.

Query 15: „We analysed glycosite locations relative to protein topology and known functional features, and found that while the majority of sites were distributed among regions of the peptide hormones typically referred to as address and message regions” – delete „while”

Response 15: Agree.

Action 15: Fixed.

Query 16: p. 9. „In some cases the intact glycopeptide hormones were still detectable even after overnight incubations.” – (First, this sentence is included twice.)

Action 16: Fixed.

Query 17: Please explain. What is „intact” here? I guess the authors refer to the peptide decorated with the largest glycan. However, these are not necessarily the native hormones.

Response 17: “Intact” refers to the non-degraded, full length mature peptide or glycopeptide. The full length mature peptide hormone is presumably the bioactive form, and in this assay we probe for how long the peptide hormones remains full length without or with different glycan structures attached. Intact does not refer to the entire “population” of full length peptide hormone, that obviously does not stay intact in many cases.

Action 17: We have modified the sentence (p. 10) as follows:

“In some cases, the intact, non-degraded glycopeptide hormones were still detectable even after overnight incubations. The protective effects seemed to be most profound with the larger glycan sialyl-T glycoforms.”

Query 18: I have some further problems with the stability studies and I should have noticed them in the first round. Nonetheless, I would like to know why the specified glycoforms were used for the stability assays. Secretin was detected glycosylated on T7, S8 and/or S11, the peptide used for the study was singly glycosylated. More importantly, the GLP-1 glycoform tested is doubly glycosylated (T5 and T7), while the authors identified only the singly glycosylated version. How relevant are these results for the native hormones?

Response 18: We identified glycosylated secretin 16 times across all dataset (9x with unambiguous identification, 2x S8, 1x T7 and S11 and 4x T7). Thus, T7 was the most abundantly identified glycoform of secretin in this dataset, and also conserved in all family members and we selected this variant for the degradation assays.

Regarding GLP-1, the glycopeptide analysed is not “doubly-glycosylated”, the chemoenzymatic production of GLP-1 resulted in a mixture of monoglycosylated GLP-1^{Thr5} and monoglycosylated GLP-1^{Thr7} approximately 1:1 and as stated in the methods section “a mixture of GLP-1^{Thr5} and GLP-1^{Thr7} was used” in the degradation assays due to limited amount of material.

Action 18: We have modified Supplementary Figure 8 and legend to stress that we are analyzing a mixture of GLP-1^{Thr5} and GLP-1^{Thr7}.

Query 19: Minor issue: the experimental part seems to be incomplete. While stability assay results are presented for PYY, the „purified glycopeptides” part on p. 19. does not indicate chemical or chemoenzymatic synthesis of the ST glycoform.

Action 19: We have modified the text on p.20 to clarify this point:

“A fraction of all T peptides was chemoenzymatically elongated to the ST form using purified ST3Gal1 (50 mM MES pH 6.5, 2 mM CMP-NeuAc (Sigma), 40 µg/mL enzyme, 1 mg/mL peptide, 37°C overnight)⁸³”

Query 20: Connecting to these studies, in the discussion the authors say (P. 12.): „The most prominent effects were observed with the naturally occurring sialylated ST O-glycan structure, where e.g. PYY^{Thr32} and GAL^{Ser23} remained intact even after 24 hrs incubation with neprilysin.” Suppl. Fig 8b clearly shows that both GAL and PYY undergo some degradation and does NOT stay intact after 24-h neprilysin treatment. Please correct.

Response 20: We disagree. As stated in reviewer 1, response 17, “intact” refers to the non-degraded full length peptide hormone. In both mentioned cases of ST-GAL and ST-PYY, the peak for the full length (“intact”) peptide hormone is still readily detectable after 24h. When comparing to the non-glycosylated variants, the peak corresponding to the full length (“intact”) peptide hormone is not detectable at all already 2h into the assay. Based on the observations that intact ST-PYY and ST-GAL remains detectable after 24h where non-glycosylated variants do not, we conclude that the protective effect of the glycans is very prominent.

Action 20: To help clarify further we have modified the text as follows:

“The most prominent effects were observed with the naturally occurring sialylated ST O-glycan structure, where e.g. full length PYY^{Thr32} and GAL^{Ser23} remained readily detectable after 24 hrs incubation with neprilysin”

Query 21: p. 10. The Discussion part starts with the following sentence: „Here, we developed a strategy to explore O-glycosylation on peptide hormones, and demonstrate that almost 33 % of all known peptide

hormones may carry O-glycans.” I think this sentence need rewording as pointed out by Referee 3 in query 1 and stick to the species/organs investigated.

Action 21: We agree and have modified the text to clarify which species were analysed in this study as follows:

“Here, we developed a strategy to explore O-glycosylation on peptide hormones, and demonstrate that almost 33 % of all known human peptide hormones may carry O-glycans in four mammalian species investigated”

Query 22: p. 10. line 283: correct „homones” to hormones

Action: Fixed.

Query 22: p. 19. line 563: replace „over night” with overnight

Action 22: Fixed.

Reviewer #2 (Remarks to the Author):

The authors have addressed the majority of concerns raised in the initial review. A few additional concerns remain:

Query 1. Page 8, lines 222-223 do not make sense and need to be revised: “When treating with OperATOR the in addition to the upper band lost intensity, a band of approximately 4-5 KDa appeared (Supplementary Fig. 6a).” Note that Supplementary is mis-spelled.

Action 1: Fixed, thank you.

Query 2. The legend for new Supplementary Figure 6b is also confusing and needs to be revised: “The theoretical mass-change by cleavage at T-glycosylated Ser3 will show approximately same mass as full length non-glycosylated proNPY (+0.1kDa) due to the size of the glycan on the remaining antibody-recognized fragment.”

Action 2: We have revised the text for Supplementary Figure 6 as follows:

“Supplementary Figure 6. Schematic illustration of pro-NPY indicating antibody recognition site and theoretical masses (KDa) of the fragments produced by OperATOR cleavage at the N-terminal site of O-glycosites identified in this study (CPON, Ser^{68/69}, Thr⁸³ and Thr⁸⁹ and NPY, Tyr²⁹, Ser³¹ and Thr⁶⁰ (Mature sites 1, 3 and 32) of which only Thr⁶⁰ is conserved among the three family members (NPY, PYY, PPY). Both glycosylation of Thr⁶⁰ in NPY and Ser^{68/69} in CPON would result in a fragment of approximately 4-5 KDa after OperATOR digestion.”

Query 3: The revised legend for Figure 1b refers to “supplementary data 3”, but the Supplementary Data Excel Spreadsheet only has tabs for Supplementary Data 1, 2, and 4.

Response 3: The Supplementary Data 3 was provided as a separate PDF file.

Action 3: We have now inserted the PDF into the Excel spreadsheet as new Supplementary data 4.

Query 4: The title for new Supplementary Figure 2 does not make sense and needs to be revised: “illustrates selected peptide hormone precursors their identified O-glycosylation sites relative to the proteolytic activation sites of peptide hormones.”

Response 4: Thank you.

Action 4: We have now added a title for Supplementary Figure 2 as follows:

"Glycosite location in relation to PC processing sites."

Reviewer #3 (Remarks to the Author):

Faced with lengthy, but very positive, critiques from all three reviewers, the authors made multiple modifications to their manuscript, making it much more accessible to the diverse audience that should benefit from the information provided. The extensive O-glycosylation of peptide hormones documented by the authors will have a profound effect on subsequent studies, on the development of peptide-based pharmaceuticals and on clinical assays for peptide hormones. The scope of the study is large, and the inclusion of structural studies, half-life studies and receptor activation studies reinforces the importance of the observations made.

That said, there are many little details that still need to be fixed. I have noted quite a few, but surely did not identify all of the singular/plural errors and repeated words. This work is too good to have it compromised by easily fixed errors.

Suggestions:

Query 1. Lines 67-68 – a few peptide hormones (e.g. calcitonin and ANP) are not "stored in secretory vesicles awaiting" – just say "most peptide hormones are stored"

Action 1: Corrected, thank you.

Query 2. In modifying their discussion of stoichiometry, the authors made this section very long. Since OperATOR was of no use on the TCA-precipitated sample, why not show only neuraminidase and move the OperATOR data for HEK293 produced proNPY to Fig.4? The "reduced intensity" argument is less than compelling and should just be deleted. Much of what appears in Results could move into the Figure legend, keeping the focus on stoichiometry.

Reply 2: Thank you for this constructive suggestion.

Action 2: We have modified Figure 4f to only show the neuraminidase produced Mw shift in STC-1. We have also moved Supplementary Figure 6a with HEK293 6E to Figure 4 (panel "g") as suggested.

The figure legend in Fig. 4 now reads:

f) & g) Western blot using an antibody recognizing pro- and mature NPY. **f)** Conditioned media from proNPY-transfected STC-1 cells was concentrated using TCA precipitation and analysed with and without neuraminidase treatment (Neu). **g)** Conditioned media from proNPY transfected HEK293 6E cells were analysed with and

without neuraminidase (Neu), OPERATOR treatment or both in combination. Both western blot experiments were repeated three times with similar results.

Query 3. In Fig.4h there are 8 test peptides/proteases – in 7 of 8 cases, the data for Tn and T are identical. The data for ST differ from the data for T in 7 of 8 cases. The final sentence of this section should not group the protective effects of the T- and sialyl-T isoforms together; it is only the sialyl-T isoform that has a more profound effect. This comment is relevant to line 309 in the Discussion, where the authors state that “the size of the O-glycan correlated positively with the observed reduction in potency (Fig.4)” – this statement should be modified to more accurately reflect the data presented.

Response 3: We agree.

Action 3: We have revised the results and discussion text related to the stability assays as follows:

” In agreement with this, we found that O-glycans at specific well conserved amino acid positions in VIP, GCG, GLP-1, NPY, PYY and PPY resulted in lower receptor potency *in vitro*. For GCG and GLP1 the size of the O-glycan correlated positively with the observed reduction in potency and for VIP all glycoforms diminished activation similarly (Fig. 4). Among the NPY receptors tested, NPY2R and NPY5R were less affected by O-glycans on NPY and PYY, while O-glycans completely abolished NPY1R and NPY4R activation.”

Query 4. In discussing the half-life data in Fig.4i, the authors group GLP-1Thr7-T and GLP-1Thr7-ST together (total GLP-1Thr7) and then present two clearly different half-lives (6.4 min and 22.7 min, respectively) – for this sentence to be understandable, the two glycoforms need to be specified, not “total glycoforms”.

Action 4: Fixed, thank you.

Query 5: Lines 333-334 – the two peptidomics studies (“both”) referred to here are not clear – please clarify.

Action 5: The sentence was redundant and has been removed.

Query 6: The widespread occurrence of O-glycosylation may also alter the ability of widely used clinical assays to detect these various peptide hormones, leading to discrepancies between laboratory data and clinical observations; this aspect of extensive O-glycosylation might be worth mentioning.

Response 6: Thank you for pointing this out.

Action 6: We have added the following sentence to the Discussion:

“Peptide hormones are frequently detected in clinical immunoassays and the presence of O-glycans on peptide hormones may mask antibody epitopes, which leads to underestimated concentrations as previously demonstrated for NT-proBNP. Thus, the atlas may also serve as a guide in the strategic design of biomarker assays.”

Minor edits

Line 35 – “peptide hormones”, not “peptides hormones”. Fixed.

Line 72 – “concentration ... has challenged”, not “have challenged”. Fixed.

Lines 139-141 – This is not a sentence; delete “while” on line 140? Fixed.

Fig.2 line 969 – Lettering is cut off. This legend is not well written. We have reworded the legend of Fig.2.

Line 970 – “illustrate that 27” – “illustrated that 27”? Fixed.

Line 974 – “sites are also predicted glycosylated” – “sites are also predicted glycosylation sites”? Fixed.

Line 975 – the graph does not show the “proportion of identified sites”, it shows the number of sites. Fixed.

Line 975 – “NetOGlyc 4.0 do not predict” – “NetOGlyc 4.0 does not predict”. Fixed.

Line 185 – “EC50-value of 200 nM EC50 compared” – should likely be “EC50-value of 200 nM compared”. Fixed.

Line 192-193 – sialic acids – plural; should be singular to go with “its strong negative charge”. Fixed.

Line 217 – “induced by neuraminidase” should be “produced by neuraminidase”. Fixed.

Lines 263-264 – The sentence starting with “In some cases ...” is repeated. Fixed.

Line 326 – need to make it clear that superscript P (XP) and superscript G (XG) indicate phosphorylation and glycosylation, respectively. Fixed.

Line 345 – “capping of glycans are needed” – should be “is needed”. Fixed.

Lines 426-427 – medium/media – the term is repeated when it should not be. Fixed.

Line 530 – Genewiz should be capitalized and a website or location given. Fixed.

Line 538 – polysciences – presumably a company with a location? Fixed.

Line 563 – Cell signaling is Cell Signaling Technologies. Fixed.

REVIEWERS' COMMENTS:

Reviewer #1 (Remarks to the Author):

The authors have answered all my questions, I suggest the acceptance of the manuscript.

My only remark is that I still think the "nongly" labels in Figure 4 are misleading, the readers might presume that the corresponding gel-separated protein(s) is/are nonglycosylated.

Point-by-point response to reviewers comment.

Reviewer #1 (Remarks to the Author):

The authors have answered all my questions, I suggest the acceptance of the manuscript.

Query1: My only remark is that I still thing the "nongly" labels in Figure 4 are misleading, the readers might presume that the corresponding gel-separated protein(s) is/are nonglycosylated.

Action 1: We have changed the labels in Figure 4 (now Figure 5) to “proteoform 1 and 2” for STC-1 and “proteoform 1, 2 and 3” for HEK produced NPY.